# Scrutinized lipid utilization disrupts Amphotericin-B responsiveness in clinical isolates of *Leishmania donovani*

**Supratim Pradhan[1], Dhruba Dhar[1], Debolina Manna[1], Shubhangi Chakraborty[1], Arkapriya Bhattacharyya[1], Khushi Chauhan[1], Rimi Mukherjee[2], Abhik Sen[2], Krishna Pandey[2], Soumen Das[1], Budhaditya Mukherjee[1]\***

[1]School of Medical Science and Technology, Indian Institute of Technology Kharagpur, Kharagpur, India; [2]ICMR-Rajendra Memorial Research Institute of Medical Sciences, Patna, India

## eLife Assessment

This **important** study investigates the propensity of the intravacuolar pathogen, Leishmania, to scavenge lipids which it utilizes for its accelerated growth within macrophages. The authors present **compelling** evidence that supports this hypothesis, although the genetic basis for the parasite's requirement for lipids remains unresolved. The study adds to other work that has implicated pathogen-derived processes in the selective recruitment of vesicles to the pathogen-containing vacuole, based on the content of the cargo.

**\*For correspondence:**
aditya26884@gmail.com

**Abstract** The management of *Leishmania donovani* (LD), responsible for fatal visceral leishmaniasis (VL), faces increasing challenges due to rising drug unresponsiveness, leading to increasing treatment failures. While hypolipidemia characterizes VL, LD, a cholesterol auxotroph, relies on host lipid scavenging for its intracellular survival. The aggressive pathology, in terms of increased organ parasite load, observed in hosts infected with antimony-unresponsive-LD (LD-R) as compared to their sensitive counterparts (LD-S), highlights LD-R's heightened reliance on host lipids. Here, we report that LD-R-infection in mice promotes fluid-phase endocytosis in the host macrophages, selectively accumulating neutral lipids while excluding oxidized-low-density lipoprotein (LDL). LD-R enhances the fusion of endocytosed LDL-vesicles with its phagolysosomal membrane and inhibits cholesterol mobilization from these vesicles by suppressing NPC-1. This provides LD-R amastigotes with excess lipids, supporting their rapid proliferation and membrane synthesis. This excess LDL-influx leads to an eventual accumulation of neutral lipid droplets around LD-R amastigotes, thereby increasing their unresponsiveness toward Amphotericin-B, a second-line amphiphilic antileishmanial. Notably, VL patients showing relapse with Amphotericin-B treatment exhibited significantly lower serum LDL and cholesterol than cured cases. Treatment with Aspirin, a lipid droplet blocker, reduced lipid droplets around LD-R amastigotes, restoring Amphotericin-B responsiveness.

## Introduction

Antimony, which was mainstay for treating fatal visceral leishmaniasis (VL), has been withdrawn from Indian Subcontinent for more than a decade now due to emergence of resistance. Despite this, recent clinical LD isolates remain unresponsive to antimony (LD-R), indicating persistent phenotype (***Dumetz et al., 2018***). Notably, LD-R also exhibits more aggressive pathology in clinical and experimental infection featuring increased organ parasite burden as compared to LD-S-infection (***Mukhopadhyay***

*et al., 2011*; *Mukherjee et al., 2013*; *Verma et al., 2010*; *Thakur et al., 2003*). Increased metacyclo-genesis among LD-R was initially thought to be the primary factor contributing to higher infectivity, resulting in organ parasite overload (*Ouakad et al., 2011*). However, later studies revealed that LD-R promastigotes have a greater replicative potential than LD-S, possibly offering a selective intracellular survival advantage (*Vanaerschot et al., 2011*; *Mukherjee et al., 2021*). This implies LD-R might be more adept at acquiring and metabolizing host-derived nutrients within their parasitophorous vacuole (PV), facilitating their heightened intracellular proliferation. Lipids are crucial carbon sources for energy in proliferating intracellular pathogens. Several intracellular bacteria, protozoans, and viruses utilizes cellular lipid droplets and plasma lipoproteins for entry, energy acquisition, and biomembrane synthesis, demonstrating intricate interactions between intracellular pathogens and host lipid metabolism (*Feingold and Grunfeld, 2012*; *Sonda and Hehl, 2006*). *Leishmania* parasites are choles-terol auxotrophs (*Goad et al., 1984*), and need (1) host membrane cholesterol for successful attach-ment and infection (*Pucadyil et al., 2004*). (2) Intracellular leishmania amastigotes replicating within membrane-bound PVs, accumulates cholesterol and lipid droplets (*Sood et al., 2024*). (3) *Leishmania* amastigotes isolated from infected host have been shown to have increased cholesterol content in them (*Bouazizi-Ben Messaoud et al., 2017*), all these clearly suggest replicating *Leishmania* amas-tigotes might co-opt host-derived lipids to facilitate their intracellular proliferation and membrane synthesis. In fact, hypocholesteremia is a hallmark of VL, with intracellular LD depleting host choles-terol and lipoproteins (*Lal et al., 2007*; *Pradhan et al., 2021*). However, the mechanism by which PV-bound LD amastigotes acquire this lipids, and how variations in lipid type, source, and utilization dynamics affect disease pathology and drug responsiveness, remains unclear.

So far, drug responsiveness in leishmaniasis has mostly been studied through in vitro selection of drug-resistant promastigotes, with limited focus on the drug susceptibility profiles of circulating clin-ical strains of *Leishmania* (*Douanne et al., 2022*; *Légaré et al., 2001*; *Gourbal et al., 2004*). However, growing evidences indicate that drug responsiveness and treatment failure in VL are multifactorial, influenced by the host's immune-metabolic status (*Mukherjee et al., 2013*; *Ghosh et al., 2012*). Importantly, recent reports have also linked emergence of drug unresponsiveness among intracellular pathogens with their ability to modulate host lipid metabolism (*Sen et al., 2011*; *Toledo and Benach, 2016*). For instance, increased lipid accumulation can hinder drug penetration by trapping lipophilic drugs, fostering resistance (*Chang et al., 2015*; *Cao, 2019*; *Day et al., 2024*). VL treatment in recent years have seen a rise in treatment failure against Amphotericin-B (Amp-B), a second-line lipophilic drug (*Mukhopadhyay et al., 2011*), making it critical to investigate these failures in the context of lipid utilization during LD infection.

Using clinical antimony-unresponsive-LD strains isolated from patients this study elucidates how LD-R-infection leads to an increased uptake of low-density lipoprotein (LDL) leading to eventual accu-mulation of neutral lipids droplets within their PV. In vitro infection of liver-resident Kupffer cells (KCs), primary sites for cholesterol synthesis and LD amastigote residence, demonstrated that LD-R-infection facilitates fluid-phase endocytosis, increasing LDL-vesicle uptake in infected KCs cells. LD-R-PV exhibits a higher propensity to fuse with endocytosed LDL-vesicles, coupled with the inhibition of cholesterol recycling from the vesicles back to the plasma membrane of infected MΦs. Our obser-vation thus explains the heightened energy source and cholesterol availability for PV and daughter membrane synthesis to accommodate higher number of LD-R amastigotes observed in clinical infec-tion. We also observed a suppressed expression of MSR-1, a scavenger receptor of oxidized-LDL (ox-LDL) (*Sheng et al., 2022*), in LD-R infected murine liver, suggesting LD-R-infection limits the uptake of ox-LDL, reducing inflammatory responses. Furthermore, our findings indicate that lipid droplet accumulation around LD-R amastigotes contributes to Amp-B unresponsiveness, explaining recent reports of Amp-B cross-resistance in clinical LD-R-isolates from India (*Purkait et al., 2012*; *Srivastava, 2011*). Significant low lipid profiles in VL patients experiencing treatment failure against liposomal Amp-B, combined with complete clearance of LD-R amastigotes using a combination treatment of Aspirin (a lipid droplet formation blocker) and Amp-B, confirmed critical role of assimilating host lipids in developing Amp-B unresponsiveness among rapidly replicating LD amastigotes exhibiting primary unresponsiveness toward antimony. This study identifies sequence of events that enables LD-R amas-tigotes to redirect host nutrients for their own aggravated sustenance, and how this preferential lipid utilization might equip them with unresponsiveness against Amp-B contributing to the gradual spread of cross-resistance among clinical LD-R-isolates.

## Results

### LD-R-infection caused severe dyslipidemia as compared to LD-S-infection

Intracellular amastigotes of *Leishmania* parasites exhibit lipid auxotrophy (*Goad et al., 1984*), with dyslipidemia being a hallmark for leishmaniasis (*Lal et al., 2007*), including VL patients. Previously, LD amastigote load in splenic aspirates of VL patients have been shown to exhibit an inverse correlation with serum cholesterol level (*Ghosh et al., 2011*). Since infection with antimony-unresponsive-LD (LD-R) has been previously reported to exhibit higher amastigote load in spleen and liver of infected host, as compared to infection with their responsive counterparts (LD-S) (*Vanaerschot et al., 2011*; *Haldar et al., 2010*), we wanted to investigate possible extent of dyslipidaemia in response to LD-R and LD-S-infection and how it might affect disease pathology. Mice infected with equal number of sorted metacyclics (*Figure 1—figure supplement 1A*) of natural clinical LD-isolates with confirmed LD-R phenotype, MHOM/IN/09/BHU575/0 (LD-R[1]) and MHOM/IN/10/BHU814/1 (LD-R[2]), showed a significantly lower serum levels of LDL, HDL, and cholesterol as compared to MHOM/IN/83/AG83 (LD-S[1]) and MHOM/IN/80/DD8 (LD-S[2]), infected mice (*Figure 1A*). Liver serves as a primary site for lipid synthesis and liver-resident MΦs (KCs) are one of the primary sites harboring LD amastigotes (*Beattie et al., 2010*). Therefore, we decided to test if KCs can be established as an in vitro model for LD-infection to investigate the possible link of host lipid utilization with respect to aggressive pathology shown by LD-R. Purified KCs (*Figure 1—figure supplement 1B*) infected with equal number of sorted metacyclics for LD-S and LD-R-isolates as before displayed no significant difference in their initial attachment to host membrane (*Figure 1B* and *Video 1A, B*). However, although initial number of intracellular LD-S and LD-R amastigotes 4 hr post infection (p.i) showed no difference, a significantly higher number of LD-R amastigotes were observed in 24 hr infected KC (*Figure 1Ci, ii*). Interestingly, as compared to a significant and sharp rise in the number of intracellular amastigotes between 4 and 24 hr post infected KC in response to LD-R-infection, the number of intracellular amastigotes although increased significantly did not doubled from 24 to 48 hr p.i. suggesting exponential LD amastigote replication between 4 and 24 hr time frame and slowing down after that (*Figure 1Ci, ii*). Moreover, it was also noticed that at 72 hr p.i. a notable number of infected KC began detaching from the wells with extracellular amastigotes probably egressing out from the infected KCs (*Video 2*). Thus, 24 hr time point was selected to conduct all further infection studies involving KCs. Parallel infection performed in peritoneal exudate macrophages (PECs), which serves as a long-standing model for in vitro LD-infection, under identical experimental conditions also resulted in similar observation with LD-R strains showing significantly higher amastigote load at 24 hr p.i. than LD-S strains (*Figure 1—figure supplement 1Ci, ii, iii*), as previously reported by multiple groups (*Dumetz et al., 2018*; *Verma et al., 2010*; *Ouakad et al., 2011*). Moreover, KCs infected with LD-S or LD-R amastigotes isolated from 28 days infected mouse spleens, also resulted in significantly higher parasite load for LD-R (*Figure 1—figure supplement 1D*), clearly indicating that higher parasite load is a conserved LD-R phenotype which does not depends on initial infectivity or MΦ origin. Notably, infected KCs had a slightly higher inherent amastigote load compared to infected PECs for both LD-S and LD-R-infections, with no significant difference in parasite load within the two independent LD-S or LD-R strains (*Figure 1C*, *Figure 1—figure supplement 1C, D*). Hence, BHU575 was selected as representative LD-R and AG83 as representative LD-S to conduct all further comparisons with the inclusion of other representative strains as and when mentioned. Higher intracellular amastigote load in response to LD-R-infection was further confirmed by imaging LD-S and LD-R-PV at 4 and 24 hr infected KCs (*Figure 1D*). Comparative analysis revealed significantly higher number of LD-R-PVs as compared to LD-S-PVs at 24 hr p.i., although this difference was not observed at 4 hr p.i. (*Figure 1Di, Dii*). Interestingly, 3D image reconstruction for LD-R-PV at 24 hr p.i., presented it as a larger fused structure as compared to smaller individual segmented appearance of LD-S-PV (*Figure 1Diii*).

### Lipid deprivation is detrimental for LD-R while inherent hypercholesterolemia facilitates its proliferation

To determine if extracellular lipids are absolutely essential for proliferation of LD-R amastigotes, LD-S and LD-R-infections were performed in high, low, or lipid-free medium. Reduction in extracellular lipids severely compromised proliferation of LD-R but not LD-S amastigotes in infected KCs, confirming

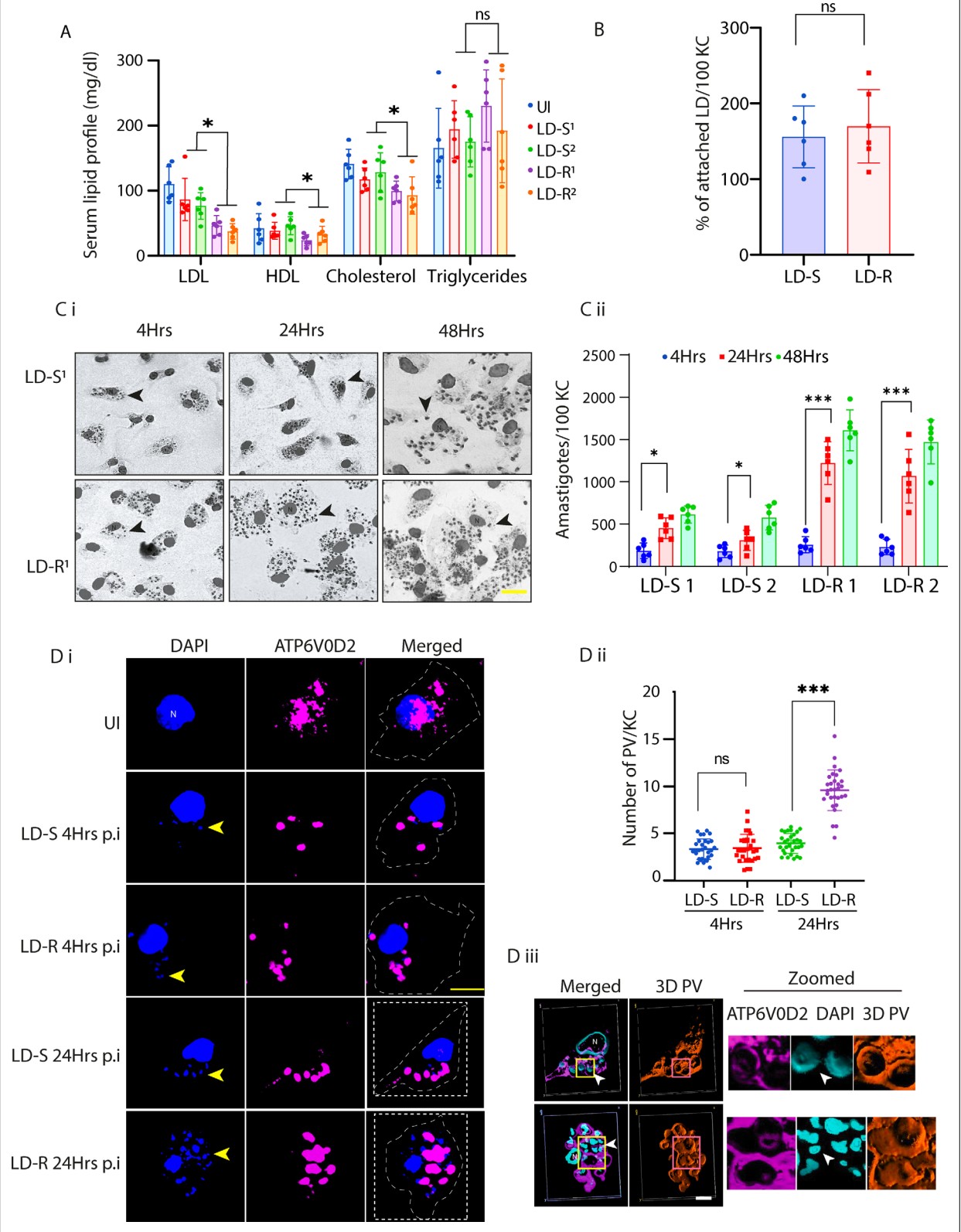

**Figure 1.** Severe dyslipidemia is linked with higher intracellular proliferation of LD-R amastigotes. (**A**) Serum lipid profile of 28 days LD-infected Balb/c mice ($N$ = 5). Equal number of sorted metacyclics promastigotes of two representative LD-S (LD-S[1], LD-S[2]) and LD-R (LD-R[1], LD-R[2]) strains were used to perform independent infection. (**B**) Number of attached LDs on Kupffer cell (KC) surface 4 hrs p.i, data represent ($N$ = 3) for two LD-S and two LD-R strains. (**C**) (**i**) Representative Giemsa stain images of LD-S[1] and LD-R[1]-infected KC at 4, 24, and 48 hr p.i. Black arrow represents LD amastigotes. $N$

*Figure 1 continued on next page*

*Figure 1 continued*

denotes host nucleus. Scale bar: 20 µM. Giemsa Images are represented in gray scale to clearly represent LD nucleus (black arrow). (**C**) (ii) Intracellular amastigote count for 4, 24, and 48 hr p.i. Each dot represents count from 100 infected KC (*N* = 6). (**D**) (i) LD-infected-KC imaged to visualize the parasitophorous vacuole (PV) at 4 and 24 hr p.i. PV marked by ATP6V0D2 (magenta), KC and LD nucleus (blue), Scale bar: 5 µm. Yellow arrow represents LD amastigotes. In the merged image, white dotted line marks KC's periphery, and host nucleus is represented with *N*. (**D**) (ii) PV counts measured from 30 infected KCs. (**D**) (iii) Confocal 3D reconstruction illustrating the spatial distribution of PVs in Kupffer cells (KCs) infected for 24 hr. ATP6V0D2, a lysosomal vacuolar ATPase subunit, is visualized in magenta, while the nucleus is depicted in cyan. The final panel highlights PV structural grooves outlined in red solid lines, with intracellular *Leishmania donovani* (LD) amastigotes indicated by white arrows. Higher magnification of (**D**) further emphasizes the increased abundance of PVs in LD-R infected cells, suggesting enhanced intracellular replication and adaptation mechanisms of drug-resistant strains. Scale bar: 5 µM. Both yellow and magenta solid line box represents the same area of the image. *** signifies p-value <0.0001, * signifies p-value <0.05, n.s., non-significant.

The online version of this article includes the following figure supplement(s) for figure 1:

**Figure supplement 1.** Kupffer cells (KCs) as an in vitro model for LD-infection.

LD-R's dependency on external lipid availability (*Figure 2Ai, ii*). Alternatively, infection performed in KCs isolated from ApoE-knockout (KO)-mice (*Apoe⁻/⁻*) with inherent diet-induced hyperlipidemia (*Hinder et al., 2013*), resulted in significant increase in amastigote load as compared to wild-type C57BL/6-infected-KCs, only for LD-R-infection, with LD-S-infection resulting in comparable amastigote load (*Figure 2Bi, ii*). Previous report has suggested that LD infection in hypercholesteremic *Apoe⁻/⁻* mice triggers a heightened inflammatory response at approximately 6 weeks' post-infection compared to wild-type BL/6 mice, leading to more efficient parasite clearance. This is owing to unique membrane composition of *Apoe⁻/⁻* which rectifies leishmania-mediated defective antigen presentation at a later stage of LD infection (*Ghosh et al., 2012*). Additionally, previous studies have also indicated that LD infection is well established in mice within 6–11 days post-infection in murine models (*Carter et al., 2003*). Thus to evaluate impact of initial lipid utilization on LD amastigote replication in vivo, BL/6 and diet-induced hypercholesterolemic *Apoe⁻/⁻* mice were infected with GFP expressing LD-S or LD-R promastigotes and sacrificed 11 days p.i. Interestingly, however, we observed *Apoe⁻/⁻* mice infected with GFP expressing LD-S or LD-R resulted in increased splenic parasite burden as compared to wild-type BL/6 mice at an earlier time point, 11 days p.i. (*Figure 2Ci, ii*), This observation possibly indicates a higher initial infection propensity in *Apoe⁻/⁻* mice, which might get resolved later on due to increasing T-cell activity. Our observation also showed that splenic parasite load for LD-R is significantly higher in *Apoe⁻/⁻* than BL/6 (*Figure 2Cii, iii*) again confirming that higher availability of extracellular lipids might help in increased proliferation of LD-R amastigotes.

## Quenching of host membrane cholesterol remains consistent during initial infection with LD-S and LD-R

Host lipids scavenging is essential for any intracellular pathogens exhibiting lipid auxotrophy (*Toledo and Benach, 2016*). Thus, to support their aggressive division, LD-R amastigotes should require significantly higher amount of host lipids, which in turn will allow excess supply of cholesterol for increased PV and daughter membrane biogenesis as compared to LD-S amastigotes. *Leishmania* amastigotes are lipid auxotrophs (*Goad et al.,*

**Video 1.** Phase contrast video representing attached LD promastigotes to peritoneal exudate cell (PEC) 6 hr p.i. Black arrow showing flagella of attached LD parasite. 1A represents LD-S and 1B represents LD-R-infection. Scale bar: 10 µm.

https://elifesciences.org/articles/102857/figures#video1

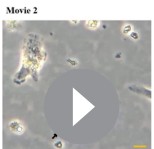

**Video 2.** Phase contrast video representing 72 hr LD infected Kupffer cells (KCs). Scale bar: 20 µm.

https://elifesciences.org/articles/102857/figures#video2

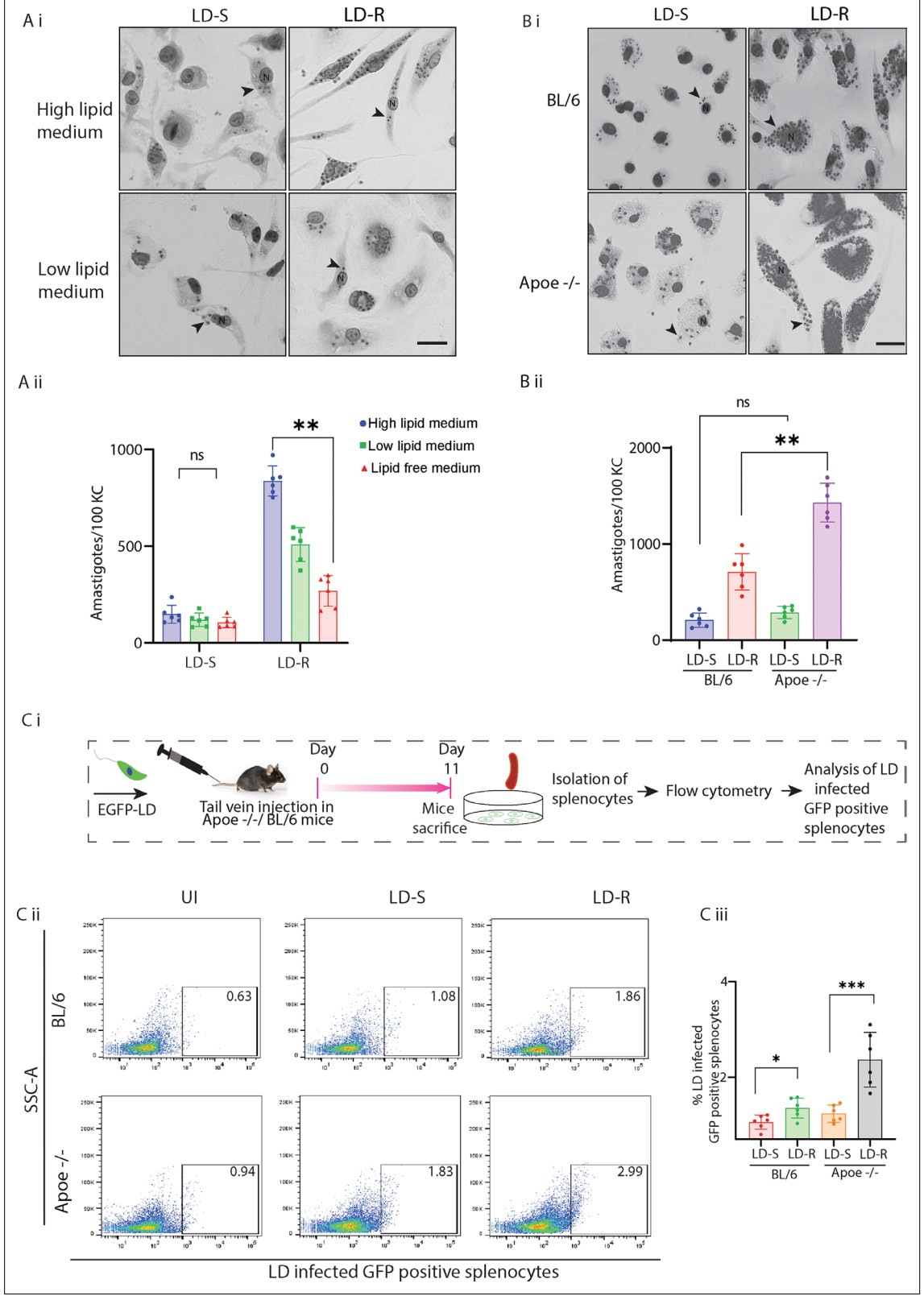

**Figure 2.** LD-R-infection results in a higher organ parasite load in hypercholesterolemic *Apoe*⁻/⁻ mice. (**A**) (i) Representative Giemsa images of LD-infected-KCs cultured in high and low lipid media. Black arrow represents LD amastigotes; N represents host nucleus. Scale bar: 20 μM. (**A**) (ii) Graph representing number of intracellular LD amastigotes in Kupffer cells (KCs) with LD infection performed either in high, low, or lipid-free media (*N* = 6). (**B**) (i) Giemsa images of LD-infected-KCs isolated from wild-type BL/6 and *Apoe*⁻/⁻ mice. Scale bar: 20 μm, N representing host nucleus. Giemsa images are

*Figure 2 continued on next page*

*Figure 2 continued*

represented in gray scale to clearly represent LD nucleus (black arrow). (**B**) (ii) Amastigote count from 100 LD-infected-KCs (*N* = 6) as in (**B**) (i). Data are presented as mean ± SEM. (**C**) (i) Schematic representation of in vivo LD-infection in BL/6 and *Apoe*⁻/⁻ mice performed with EGFP-LD-S or EGFP-LD-R. (**C**) (ii) Flow cytometry representing GFP-positive splenocytes isolated from 11 days EGFP-LD infected BL/6 and *Apoe*⁻/⁻ mice. The black box indicates % of GFP-positive splenocytes (LD-infected). (**C**) (iii) Graph representing data from six infected mice as in (**C**) (ii) are presented. *** signifies p-value <0.0001, ** signifies p-value <0.001, * signifies p-value <0.05, n.s., non-significant.

*1984*; *Bouazizi-Ben Messaoud et al., 2017*), and it has been previously reported that *L. mexicana* can acquire host membrane cholesterol during its initial entry into host MΦs (*Semini et al., 2017*). Thus, host membrane cholesterol was investigated as a possible source and potential mechanism by which LD-R acquires host-derived lipids distinctively from their LD-S counterparts. However, confocal image analysis of NBD-cholesterol labeled KCs infected with RFP expressing LD-S or LD-R revealed that both LD-S and LD-R can extract NBD-cholesterol from the KC membrane during their initial infection (4 hr p.i.) with comparable efficiency (*Figure 3—figure supplement 1Ai, ii*). This observation was further confirmed by quantification of NBD-cholesterol between freshly invaded LD-S and LD-R parasites isolated from infected KC by flow cytometry (*Figure 3—figure supplement 1Bi, ii*).

## LD-R-infection mobilizes extracellular LDL within infected KCs to support its intracellular proliferation

Since the quenching of host membrane cholesterol during initial entry of LD promastigotes does not significantly differ in LD-S- and LD-R-mediated infections (*Figure 3—figure supplement 1*), we explored other potential host lipid sources for supporting LD-R's increased intracellular division and membrane synthesis. LDL serves as the primary source of cellular lipids which helps to maintain lipid homeostasis, and gets readily metabolized to produce bioavailable cholesterol (*Vance, 2022*). Our previous observation showing a significant drop in LDL levels between LD-S and LD-R-infected mice (*Figure 1*), led us to investigate LDL as a possible source which can differentially contribute in providing excess lipids, sterols, and fatty acids to rapidly proliferating intracellular LD-R amastigotes. Live cell imaging comparing uptake of fluorescent red-LDL in EGFP-LD-S or EGFP-LD-R-infected KCs at 24 hr p.i. revealed a significantly higher red-LDL accumulation within EGFP-positive LD-R-infected KCs as compared to LD-S-infection (*Figure 3Ai*). Fluorescence plate reader-based quantification further supported this observation showing significantly higher LDL accumulation in LD-R-infected-KCs over time, which was not observed by performing infection with killed parasites (*Figure 3Aii*), indicating that, LDL-influx is mediated by replicating LD amastigotes. Confocal microscopic images at 24 hr p.i. of LD-S and LD-R-infected-KCs confirmed significantly higher LDL co-localizing with LD-R amastigotes compared to LD-S amastigotes (*Figure 3Bi, ii*).

## LD-R-infection results in a distinct metabolic shift in lipid peaks of infected host

Next, we wanted to confirm if higher LDL uptake could result in accumulation of excess lipids, particularly cholesterol, in LD-R-PV that would provide LD-R amastigotes with higher replicative capabilities and associated enhanced membrane synthesis potential. PV fractions were isolated from LD-S and LD-R-infected KCs at 24 hr p.i. using a previously established protocol (*Pessoa et al., 2019*). Following isolation, PV purity was confirmed through LAMP-1 staining which showed a significant enrichment around isolated PV in confocal microscopy (*Figure 3Ci*). Purity of isolated PV fractions was further confirmed by western blot which showed an enhanced enrichment of LAMP-1 for LD-R-PV fraction as compared to LD-S-PV fraction, while PV excluded cellular fraction showed residual LAMP-1 expression confirming the purity of the isolated PV fractions (*Figure 3Cii, iii*). Following isolation, protein concentration was measured for isolated PV fractions using the Bradford assay, and PV fractions from both LD-S- and LD-R-infected KCs were normalized accordingly. Quantitative analysis of total cholesterol using Amplex red kit (*Figure 3—figure supplement 2A*) and thin layer chromatography (TLC) (*Figure 3—figure supplement 2B*) revealed significantly higher cholesterol content in LD-R-PV as compared to LD-S-PV. Furthermore, gas chromatography–mass spectrometry (GC–MS) of LD-S and LD-R-PV under identical experimental condition identified prominent peaks for cholesterol and fatty acids in LD-R-PV which were missing for LD-S-PV (*Figure 3Di, ii*). Apart from this biochemical characterization, we also performed biophysical analysis of lipid-related peaks using

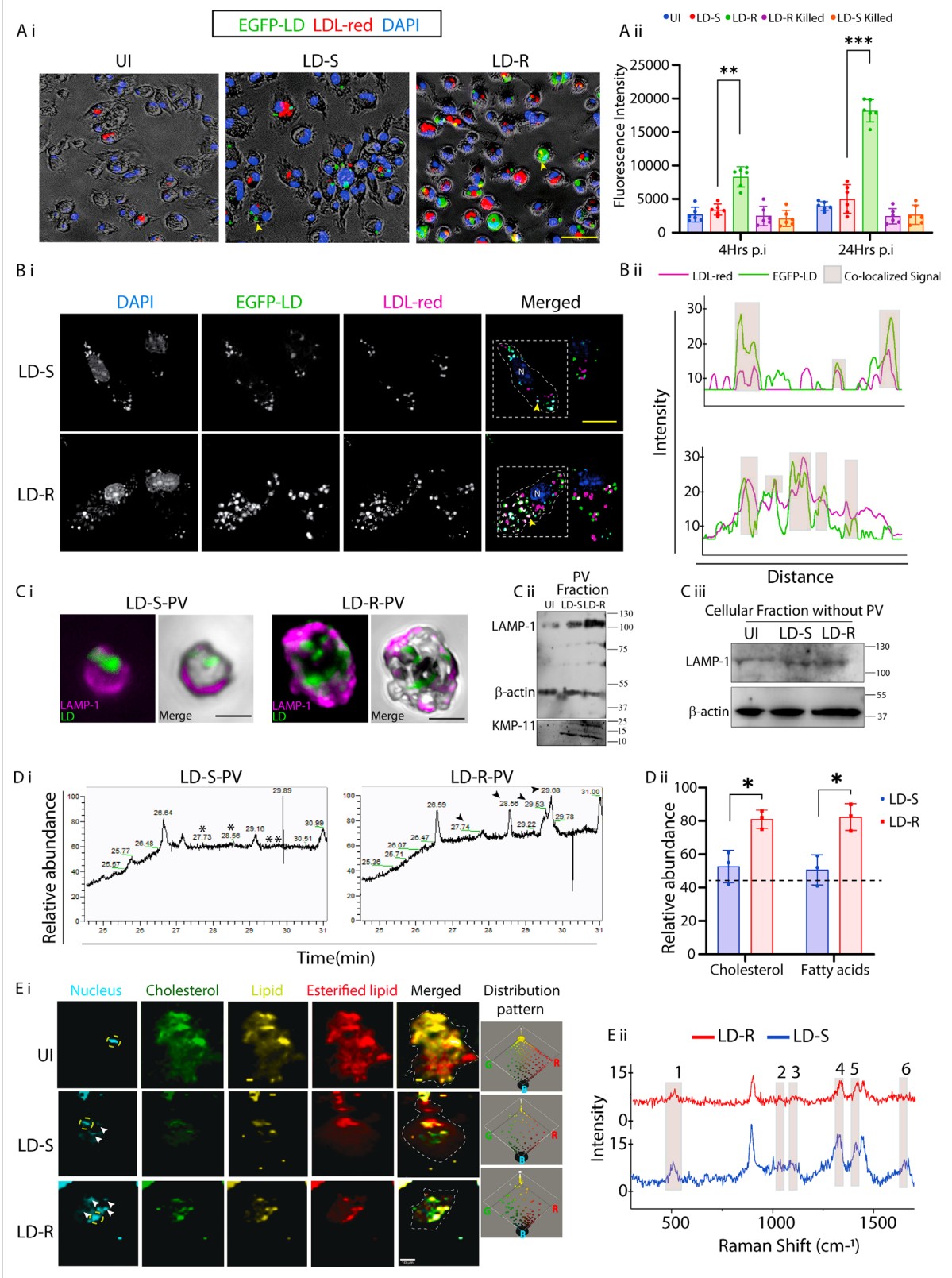

**Figure 3.** Endocytosed-low-density lipoprotein (LDL) is the primary lipid source for intracellular LD-R proliferation. (**A**) (i) Live cell images of LDL-endocytosis in 24 hr post infected Kupffer cells (KCs) with LDL (red), LD (green), and the nucleus (blue). Images were acquired directly from 96-well glass bottom plate and bright field was merged with reduced brightness and increased contrast identically for all experimental conditions. Scale bar: 40 µm. Yellow arrow represents intracellular LD. (**A**) (ii) Graph representing fluorescence plate reader-based quantification of (**A**) (i). (**B**) (i) Co-localization of LDL

*Figure 3 continued*

with the LD-amastigotes in infected KCs 24 hr p.i. Merged image showing LDL (magenta), LD (green), nucleus (blue), white dotted line in the merged image demarcates infected KC for co-localization analysis. Scale bar: 10 μm. Yellow arrow represents LD nucleus and N represents host nucleus. (**B**) (ii) Intensity plot representing co-localized signals from (**B**) (i), marked with gray transparent box. (**C**) (i) Confocal image showing the isolated LD-S and LD-R-PV, stained with LAMP1 (magenta) and LD (green). Scale bar: 2 μm. (**C**) (ii) Western blot analysis validated the purification of parasitophorous vacuole (PV) fractions, indicated by LAMP-1 positivity, while the presence of LD parasites within the PV was confirmed by KMP-11 detection. (**C**) (iii) Western blot of cellular fraction without PV showing minimal presence of LAMP1. (**D**) (i) Gas chromatography–mass spectrometry (GC–MS) lipid profile of isolated PVs from LD-infected-KCs. Retention time 27.73 corresponding to the cholesterol and 28.56 corresponding to fatty acids are represented with * for LD-S-PV and with black arrowheads for LD-R-PV. (**D**) (ii) Relative abundance of cholesterol and fatty acids of (**D**) (i). (**E**) (i) Representative confocal Raman spectroscopy image of uninfected or LD-infected-KCs 24 hr p.i., illustrating distinct lipid-related signal distribution patterns marked with pseudo-colors. Yellow dotted circle in the left most panel demarcates the host cell nucleus while white arrow marking intracellular LD nucleus. In merged panel, white dotted line demarcates cells periphery. The right most panel shows dot plot representation of lipid distribution, where each color corresponds to different lipid-related peaks with respect to blue (**B**) indicating the nucleus. (**E**) (ii) Comparative Raman spectra from LD-S and LD-R-infected-KCs. Lipid-related peaks are demarcated with shaded box, representing 1. 540–560 cm$^{-1}$ (cholesterol), 2. 1080–1090 cm$^{-1}$ (phospholipids), 3. 1270–1280 cm$^{-1}$ (triglycerides), 4. 1300–1340 cm$^{-1}$ (Amide-III bond), 5. 1440–1453 cm$^{-1}$ (fatty acids and triglycerides), and 6. 1650–1660 cm$^{-1}$ (Amide-I bond). *** signifies p-value <0.0001, ** signifies p-value <0.001, * signifies p-value <0.05, n.s., non-significant.

The online version of this article includes the following source data and figure supplement(s) for figure 3:

**Source data 1.** Source data for western blots shown in *Figure 3Cii and iii*.

**Source data 2.** Source data for western blots shown in *Figure 3Cii and iii*, with relevant bands labelled.

**Figure supplement 1.** LD-S and LD-R quenches host membrane cholesterol during initial entry.

**Figure supplement 2.** LD-R-PV have higher accumulation of cholesterol and fatty acids.

non-invasive confocal Raman spectroscopy to capture a snapshot of lipid distribution in LD-S and LD-R-infected-KCs under native conditions (*Figure 3Ei, ii*). This analysis highlighted a distinct clustering of lipid peaks representing cholesterol, triglycerides, and fatty acids (*Feuerer et al., 2021*) around LD-R amastigotes, contrasting with their minimal clustering around LD-S amastigotes which resemble uninfected-KCs more closely (*Figure 3Ei*). Raman shift analysis within the biologically active region (300–1800 cm$^{-1}$) (*Živanović et al., 2018*), further confirmed a significant difference in relative abundance for lipid-related peaks at wavelengths: 540–560 cm$^{-1}$ (cholesterol), 1080–1090 cm$^{-1}$ (phospholipids), 1270–1280 cm$^{-1}$ (triglycerides), 1300–1340 cm$^{-1}$ (Amide-III bond), 1440–1453 cm$^{-1}$ (fatty acids and triglycerides) (*Feuerer et al., 2021*; *Tiwari et al., 2020*), along with olefinic stretch of C=C double bonds at 1655 cm$^{-1}$ between LD-S and LD-R-infected-KC, at 24 hr p.i. (*Figure 3Eii* and *Figure 3—figure supplement 2Ci, ii*). We also noticed a distinct shift by plotting area under the curve from (right to left) in response to LD-R-infection (*Figure 3—figure supplement 2D*), indicative of distinctly different lipidomes (*Beattie et al., 2004*).

## LDL uptake in LD-infected-MΦs occurs through fluid-phase endocytosis

As LD-R amastigotes heavy reliance on increased host lipid acquisition for rapid proliferation became evident, we wanted to investigate molecular basis of this acquisition. Considering that MΦs primarily uptake LDL via the LDL receptor (LDLr) (*Goldstein et al., 1985*), we checked its status in response to LD-S and LD-R-infection. However, immunofluorescence analysis showed a decrease in LDLr expression in infected KCs harboring LD-amastigotes as compared to adjacent uninfected KCs, regardless of infection type (*Figure 4A*), along with no significant changes in total LDLr expression between uninfected, LD-S, and LD-R-infected-KCs, as observed from western blot (*Figure 4—figure supplement 1A*). This indicated that LDL uptake by LD-infected-KCs might be receptor independent. Our attempts to generate siRNA-mediated knock down (KD) in KCs were unsuccessful, hence to convincingly determine the role of LDLr in LD-infected-MΦs we generated LDLr-KD-PEC (*Figure 4—figure supplement 1Bi, ii*) and performed infection with LD-S or LD-R. Infecting LDLr-KD-PEC with LD-S or LD-R resulted in no significant changes in intracellular amastigote numbers for both infection types as compared to scrambled RNA control (*Figure 4—figure supplement 1C*) confirming that although LD-R relies on increased LDL-influx for heightened proliferation, this influx is LDLr independent. Apart from LDLr, MΦs can also acquire LDL through receptor-independent fluid-phase endocytosis (*Lucero et al., 2020*). To confirm this possibility, we checked the status of Cofilin, an actin modulating protein, which serves as a marker for fluid-phase endocytosis. Western blot analysis of infected KCs revealed an increase in Cofilin levels with a concurrent decrease in its phosphorylated form as LD-R-infection

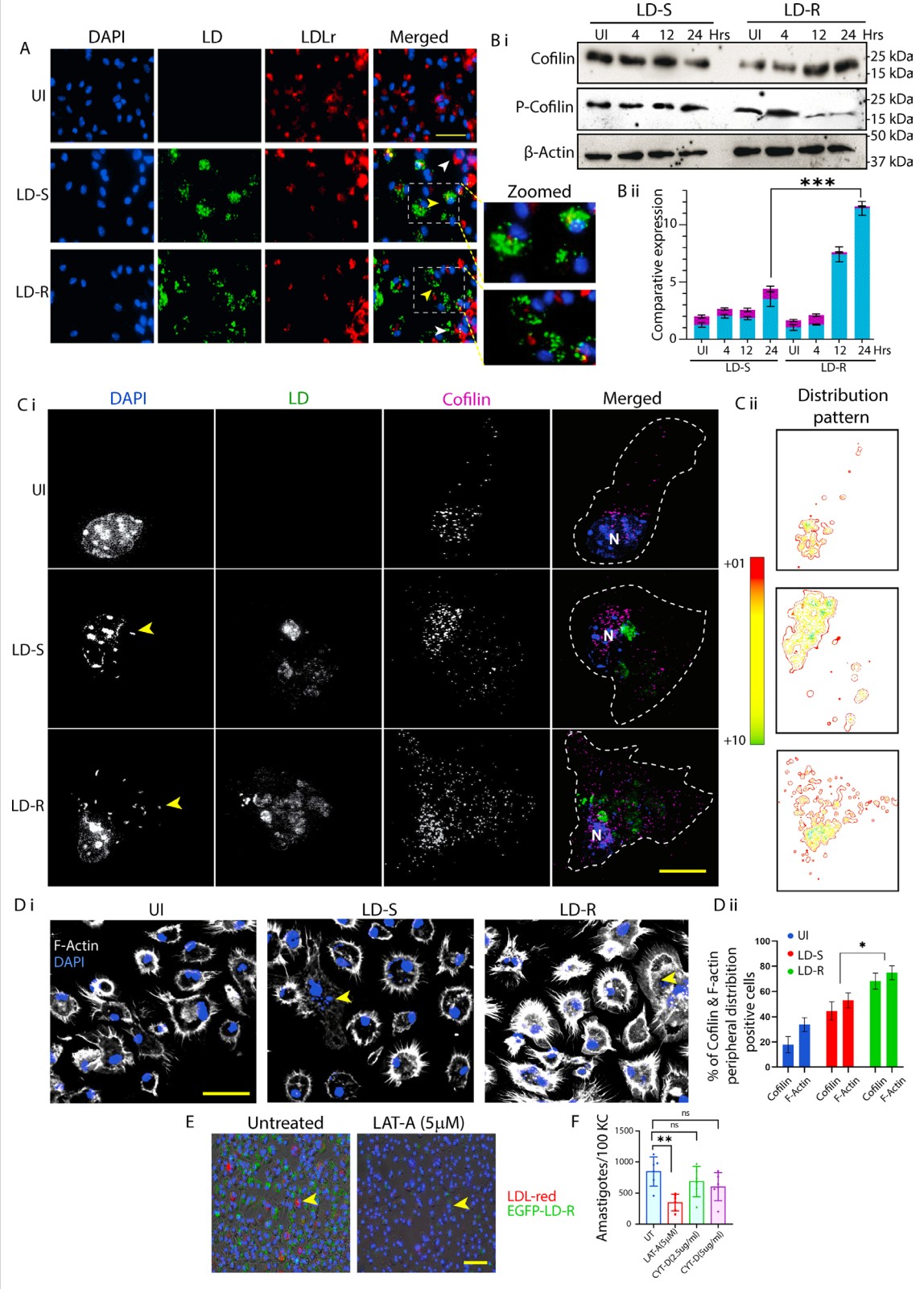

**Figure 4.** Increased low-density lipoprotein (LDL) uptake in LD-R-infected-MΦs through receptor-independent fluid-phase endocytosis. (**A**) Immunofluorescence images of LDLr expression in LD-infected-KCs. Scale bar: 40 μm. Enlarged view of A showing infected KC with low LDLr expression (yellow arrow) compared to neighboring uninfected Kupffer cell (KC; white arrow) in the merged image. (**B**) (i) Cofilin and phosphorylated-Cofilin expression by western blot in LD-infected-KCs. (**B**) (ii) Graphical representation of comparative expression of Cofilin and phosphorylated-Cofilin with

*Figure 4 continued on next page*

*Figure 4 continued*

respect to β-actin for B (i). (**C**) (i) Structural illumination microscopy (SIM) images showing the Cofilin Distribution in LD-infected-KCs. Yellow arrow represents intracellular LD, white dotted line in the merged image demarcates cell boundary. Scale bar: 5 µm. (**C**) (ii) Color spectrum representing Cofilin Distribution pattern of (**C**) (i) with green representing maximum and red representing minimum Distribution. (**D**) (i) Confocal images representing Filamentous actin (F-actin) protrusions in LD-infected-KCs. Scale bar: 20 µm, with yellow arrow representing LD amastigotes. Individual experimental condition is represented in pseudo-color, with uniform high contrast to clearly represent the F-actin protrusions. (**D**) (ii) Graph representing Cofilin and F-actin Distribution in KCs infected with LD or left uninfected. (**E**) Still images from live cell video microscopy representing co-localization of LD-R with LDL in presence or absence of Latrunculin-A (LAT-A). Scale bar: 60 µm. Yellow arrow represents LD-infected-KCs with LDL (red). (**F**) Graphical representation (*N* = 6) of amastigote count in 100 LD-R-infected-KCs treated with either LAT-A, or Cytochalasin (CYT-D) or left untreated. Data are presented as mean ± SEM. *** signifies p-value <0.0001 ** signifies p-value <0.001, * signifies p-value <0.05, n.s., non-significant.

The online version of this article includes the following source data and figure supplement(s) for figure 4:

**Source data 1.** Source data for western blots shown in *Figure 4Bi*.

**Source data 2.** Source data for western blots shown in *Figure 4Bi*, with relevant bands labelled.

**Figure supplement 1.** Latrunculin-A restricts extracellular low-density lipoprotein (LDL) influx inhibiting LD-R amastigote proliferation.

**Figure supplement 1—source data 1.** Source data for western blots shown in *Figure 4—figure supplement 1A, Bi, Di*.

**Figure supplement 1—source data 2.** Source data for western blots shown in *Figure 4—figure supplement 1A, Bi, Di*, with relevant bands labelled.

progressed, indicating active fluid-phase endocytosis (*Figure 4Bi, ii*). In contrast, LD-S-infection did not induce such changes (*Figure 4Bi, ii*). Interestingly, this decrease in Cofilin phosphorylation was also observed in LD-R-infected-PEC (*Figure 4—figure supplement 1Di, ii*), signifying that enhanced fluid-phase endocytosis may be a conserved mechanism for facilitating LDL-influx in LD-R-infected-MΦs. This enhanced fluid-phase endocytosis in response to LD-R-infection was further confirmed by structural illumination microscopy (SIM) imaging, which showed increased Cofilin turnover with dispersed distribution in LD-R-infected-KCs, leading to enhanced F-Actin rearrangement and protrusion formation (*Popow-Woźniak et al., 2012*), which was absent for uninfected or LD-S-infected-KCs (*Figure 4Ci, ii, Di, ii*). Finally, we infected the KCs with GFP expressing LD-R for 4 hr, washed and allowed the infection to proceed in presence of fluorescent red-LDL and Latrunculin-A (5 µM), a compound which specifically inhibits fluid-phase endocytosis by inducing actin depolymerization (*Lucero et al., 2020*). Real-time fluorescence tracking demonstrated that Latrunculin-A treatment not only prevented the uptake of fluorescent red-LDL but also severely impacted intracellular proliferation of LD-R amastigotes (*Video 3A, B* and *Figure 4E*). In contrast, treatment with Cytochalasin-D, which alters cellular F-actin organization but does not affect fluid-phase endocytosis (*Lucero et al., 2020*), had no effect on the intracellular proliferation of LD-R irrespective of Cytochalasin-D concentrations (2.5 and 5 µg/ml, respectively) (*Figure 4F* and *Figure 4—figure supplement 1E*).

## Endocytosed-LDL traffics and fuses with and LD-R-PV with increased proficiency

Typically, LDL-vesicles fuses with lysosomes upon entry into MΦs, leading to subsequent transfer

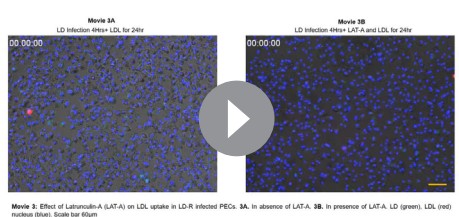

**Video 3.** Effect of Latrunculin-A (LAT-A) on low-density lipoprotein (LDL) uptake in LD-R infected peritoneal exudate cells (PECs). (3A) In absence of LAT-A. (3B) In presence of LAT-A. LD (green), LDL (red), and nucleus (blue). Scale bar: 60 µm.

https://elifesciences.org/articles/102857/figures#video3

of LDL-derived cholesterol into the endoplasmic reticulum and cytoplasm (*Urano et al., 2008*). As we have already confirmed an increased influx with higher presence of endocytosed-LDL-vesicles around LD-R amastigotes along with significantly higher accumulation of cholesterol and fatty acids in LD-R-PV as compared to LD-S-PV (*Figure 2*), we hypothesized that the mode of trafficking of endocytosed-LDL might differ between LD-S and LD-R-infection. LD-PV are modified phagolysosomes (*Pessoa et al., 2019*; *Verma et al., 2017*), and we observed an increased convergence of LDL-vesicles with LAMP-1-positive LD-R-PV even at 4 hr p.i., while such convergence was not noticed in LD-S-infected-KC (*Figure 5Ai, ii*). Structured illumination microscopy confirmed a

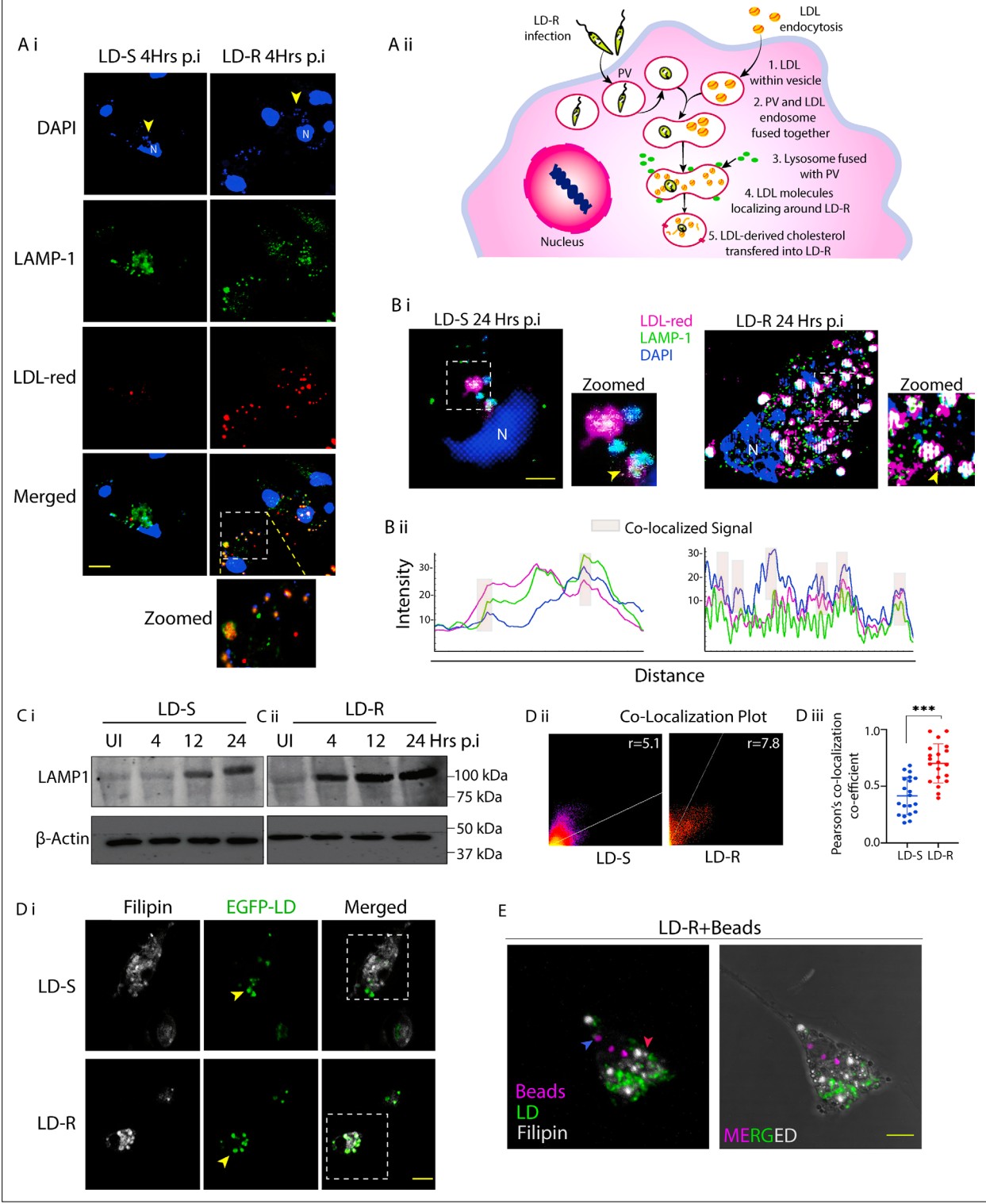

**Figure 5.** Endocytosed-low-density lipoprotein (LDL) fuses with LD-R-PV to provide cholesterol to LD-R amastigotes. (**A**) (i) Confocal images of representing LDL-vesicles, with LAMP1-positive lysosomal vesicles in LD-infected-KCs 4 hr p.i. Yellow arrow represents LD nucleus, white dotted box marks the region further cropped and zoomed to show fusion of lysosomal vesicles with LDL and LD-R. Scale bar: 10 μm. (**A**) (ii) Schematic representation showing trafficking of LDL-vesicle toward the LD-R-PV. (**B**) (i) Structural illumination microscopy (SIM) images representing convergence of LDL-vesicles with LD amastigotes and LAMP1 in 24 hr with LD-infected-KCs. White dotted box marks the region further cropped and zoomed to clearly show the convergence of LD amastigotes with LDL and LAMP-1 (white vesicles shown by yellow arrow). Scale bar: 5 μm. (**B**) (ii) Intensity plot representing of co-localization signals for (**B**) (i). (**C**) Western blot analysis showing LAMP1 expression in LD-infected-KCs from early to late hours. (**C**)

*Figure 5 continued on next page*

Figure 5 continued

(i), (ii) representing LAMP1 expression LD-S and LD-R-infected-KC (**D**) (i) Filipin staining of EGFP-LD-infected-KCs. Filipin (white) and LD (green), yellow arrow represents intracellular LD. Scale bar: 10 μm. White dotted line marked cells used to generate co-localization plot. (**D**) (ii) Co-localization signals of LD (EGFP) and Filipin represented by the dot plot. Pearson co-localization coefficient value (*r*) represented for LD-S and LD-R-infected-KC. (**D**) (iii) Graph representing Pearson's coefficient for EGFP-LD-infected-KCs (*N* = 19). (**E**) Filipin staining of EGFP-LD-R-infected KCs incubated with dextran beads (magenta) revealed minimal colocalization between the beads and Filipin, while the LD-R amastigotes within same macrophage shows a significant co-localization with Filipin. Blue arrow demarking beads with limited filipin colocalization, red arrow demarking LD-R with filipin colocalization. Scale bar: 5 μM. Statistical significance in the observed differences was determined through ANOVA utilizing GraphPad Prism software (version 9.0). \*\*\* signifies p-value <0.0001.

The online version of this article includes the following source data for figure 5:

**Source data 1.** Source data for western blots shown in *Figure 5Ci*.

**Source data 2.** Source data for western blots shown in *Figure 5Ci*, with relevant bands labelled.

complete convergence of LDL-vesicles with LD-R-PV at 24 hr p.i., while LD-S-PV showed only partial convergence with LDL-vesicles even at this late infection point (*Figure 5Bi, ii*). Interestingly, while, western blot analysis revealed minimal expression of LAMP-1 in response to LD-S-infection at 4 hr p.i., which increased considerably as the infection progressed (*Figure 5Ci*), however, a considerable amount of LAMP-1 expression was observed for LD-R-infected-KCs from 4 hr p.i., which also increased as infection progresses (*Figure 5Cii*), indicating that a higher fusion of LDL-vesicles in LD-R-PV may be accompanied by increased lysosomal biogenesis required for degrading LDL into cholesterol. Filipin staining of EGFP-LD-S and EGFP-LD-R-infected KCs at 24 hr p.i. revealed enhanced cholesterol accumulation surrounding LD-R amastigotes, as indicated by a significantly higher co-localization coefficient compared to LD-S amastigotes (*Figure 5Di–iii*). Importantly, this cholesterol enrichment was absent around dextran beads harboring LD-R amastigotes, suggesting that cholesterol accumulation is a process specifically associated with LD-R-infection. Infected KCs harboring both LD-R amastigotes and Fluorescent Latex Beads, showed a concentrated staining of cholesterol around LD-R amastigotes, with no positive cholesterol staining around internalized latex beads similar to LD-S amastigotes (*Figure 5E*).

## LD-R-infection suppress release of metabolized cholesterol from PV

Once LDL is broken down to cholesterol by lysosomal acid hydrolases, metabolized cholesterol is released back into cytosol from LDL-vesicles by two transmembrane proteins namely NPC-1 and NPC-2 (*Urano et al., 2008*). We suspected that LD-R must prevent or delay this process as this will limit cholesterol accessibility to replicating LD-R amastigotes. Analysis of available RNAseq (INRP000146) data for uninfected, LD-S and LD-R-infected-PEC (*Figure 6A*) revealed a suppression of *Npc1* expression along with its transcription factor *Srebp2* (*Garver et al., 2008*), in response to LD-R-infection. Western blot analysis of LD-S and LD-R-infected-KCs confirmed this low expression of SREBP2 and NPC-1 in LD-R-infected-KC 24 hr p.i. (*Figure 6Bi, ii*). More interestingly, confocal imaging not only confirmed this low expression of SREBP2, but even in those cases of LD-R-infected-KCs where we did observe some residual expression of SREBP2, it was entirely restricted to cytoplasm, unlike LD-S or uninfected-KCs where SREBP2 did get translocated in the nucleus (*Figure 6C*). Furthermore, as SREBP2 acts as the master regulator of cholesterol biosynthesis, its downregulation and inhibition of nuclear translocation further impacted expression of the downstream cholesterol biosynthetic enzyme HMGCR (*Figure 6D*) specifically in LD-R-infected-KCs. This again supports the RNAseq expression data (*Figure 6A*) and further confirms that LD-R-infected-KCs has a shutdown of de novo cholesterol synthesis. For NPC-1, immunofluorescence imaging comparing adjacent uninfected and infected KCs harboring LD-S or LD-R, clearly confirmed unlike LD-S, LD-R specifically suppresses NPC-1 to prevent release of LDL-derived cholesterol from fused vesicles making it accessible to amastigotes (*Figure 6E*). We confirmed this observation by performing a total internal reflection fluorescence microscopy (TIRF) which clearly showed a low plasma membrane cholesterol for LD-R-infected-KCs as compared to LD-S-infected-KCs (*Figure 6Fi*) and subsequently revalidated this by quantifying the cholesterol content in isolated membrane fraction for individual experimental set as mentioned above (*Figure 6Fii*). Finally, siRNA-mediated KD of NPC-1 in PEC (due to failure of successful KD generation in KCs), resulted in significant increase in amastigote load in response to LD-R-infection, while no such change in amastigote load was observed for LD-S-infected NPC-1-KD-PECs (*Figure 6Gi–iv*).

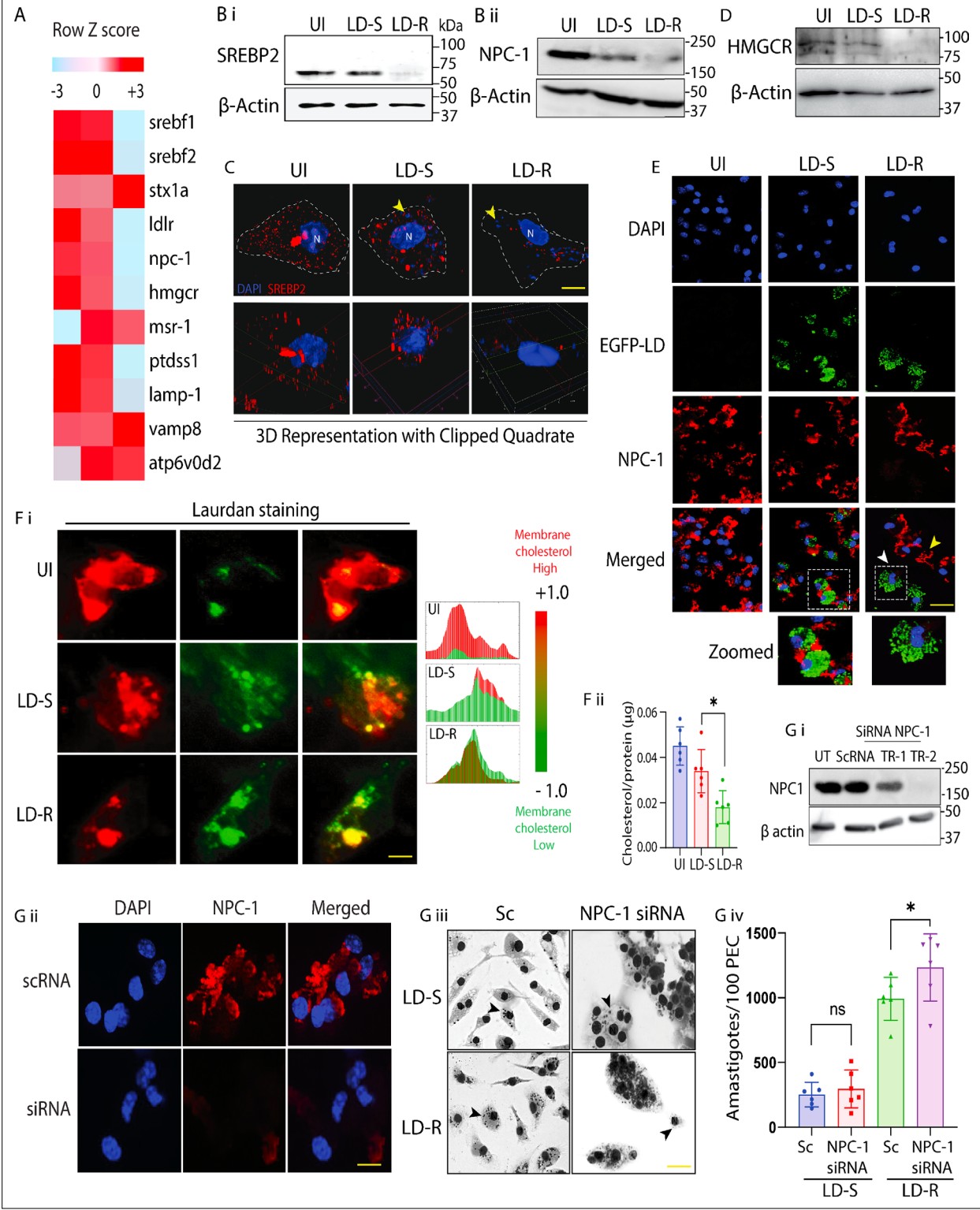

**Figure 6.** LD-R-infection suppress NPC-1 to alter cholesterol mobilization in infected MΦs. (**A**) 24 hr LD-S and LD-R-infected-PECs were subjected to RNAseq analysis keeping uninfected-PECs as control. Heat map representing average expression of differentially expressed genes related to lipid metabolism between LD-S and LD-R-infected-PEC. Scale represents median-centered counts (TPM). Row $Z$ score data represented here. Expression of RNAs (right margins) is presented as centered and 'scaled' [mean normalized $\log_2(\text{TPM} + 1)$]. (**B**) (i) Expression of SREBP2 and (**B**) (ii) NPC-1 in LD-infected and uninfected Kupffer cells (KCs) by western blot with β-actin as control. (**C**) Intracellular localization of SREBP2 in response to LD-infection in KC 24 hr p.i. White dotted line represents cell periphery with yellow arrow representing intracellular LD and N representing host nucleus. 3D image

*Figure 6 continued on next page*

*Figure 6 continued*

represented below with one quadrant clipped to confirm nuclear translocation of SREBP2. Scale bar: 5 μm. (**D**) Expression of HMGCR LD-infected and uninfected KCs by western blot with β-actin as control. (**E**) Immunofluorescence showing NPC-1 expression in LD-infected-KCs. White arrow representing NPC-1 expression in LD-R-infected-KCs and yellow arrow representing NPC-1 in adjacent uninfected-KCs in the merged panel. White dotted box marks regions further cropped and shown in the bottom panel with enlarged view (Zoomed). Scale bar: 20 μm. (**F**) (i) Total internal reflection fluorescence (TIRF) microscopic images of Laurdan stained LD-infected and uninfected KCs with two different spectra (488 and 594 nm), changes from red to green represents gel to fluid-phase transition of the host plasma membrane. Color distribution pattern corresponding to the TIRF images representing each experimental condition is presented in the right most panel. Scale bar: 5 μm. (**F**) (ii) Measurement of membrane cholesterol by Amplex red assay kit. Graphical representation showing total membrane cholesterol divided by total membrane protein. (**G**) (i) Expression of NPC-1 determined by western blot in peritoneal exudate cells (PECs) transfected with NPC-1 siRNA or scrambled (sc) siRNA. (**G**) (ii) Expression of NPC-1 determined by immunofluorescence in PECs transfected with NPC-1 siRNA or scrambled (sc) siRNA. Scale bar: 5 μm. (**G**) (iii) Giemsa-stained images of LD-infected NPC-1 knockdown-PECs with scrambled siRNA control (Sc). Scale bar: 20 μm. Images are represented in gray scale with increased contrast to represent LD nucleus (black arrow). (**G**) (iv) Graphical representation of number of intracellular LD-amastigotes as in (**F**) (ii). Results show the counting of 100 infected PEC from 6 independent replicate. * signifies p-value <0.05, n.s., non-significant.

The online version of this article includes the following source data for figure 6:

**Source data 1.** Source data for western blots shown in *Figure 6Bi, ii, D, Gi*.

**Source data 2.** Source data for western blots shown in *Figure 6Bi, ii, D, Gi*, with relevant bands labelled.

## LD-R-infection specifically excludes ox-LDL in infected murine liver to limit inflammation

The increased accumulation of LDL within MΦs typically triggers an inflammatory response (*Ho et al., 2019*). This possessed an inherent conceptual dilemma to our observation as increased inflammatory response should be detrimental for LD-R amastigotes (*Murray et al., 1982*; *Kima and Soong, 2013*). To check this, we measured inflammatory IFN-γ by co-culturing splenic T-cells isolated from LD-S or LD-R-infected mice, with LD-S and LD-R-infected-KCs, respectively (*Figure 7A*). Surprisingly, our ELISA results showed a low IFN-γ levels in the culture supernatant of LD-R-infected-KCs even after 24 hr of co-culture, while for LD-S-infection a significant increase was noted at this point (*Figure 7Bi*). Similarly, stimulation of LD-infected splenocytes obtained from LD-S and LD-R-infected *Apoe⁻/⁻* mice with specific soluble *Leishmamia* antigens (SLA) of LD-S or LD-R also confirmed this restricted IFN-γ response to LD-R-infection (*Figure 7Bii*), thus suggesting that although LD-R-infection results in increased accumulation of lipids in host MΦs, it elicits a suppressed inflammatory response as compared to LD-S. Notably, accumulation of ox-LDL, and not its neutral form has been reported to cause inflammatory response (*Ho et al., 2019*; *Govaere et al., 2022*). LD-S or LD-R-infected-KCs incubated with ox-LDL showed a significant purple positive staining of ox-LDL only for LD-S-infected-KCs (*Figure 7Ci*), with LD-R-infected-KCs seemingly inhibiting uptake of ox-LDL. Infecting KCs with higher number of LD-S or lower number of LD-R did not alter their ox-LDL uptake capabilities, clearly suggesting LD-R-infection inherently suppress ox-LDL uptake, irrespective of the number of intracellular amastigotes (*Figure 7Cii*). Although, receptor of ox-LDL, MSR-1 (*Sheng et al., 2022*), has shown a slightly lower expression in LD-R-infected-PEC in the RNAseq analysis (*Figure 6A*), however owing to reported dominant expression of *msr-1* in the liver (*Govaere et al., 2022*), we checked expression of MSR-1 in the LD-S and LD-R-infected-KCs. Confocal images and western blot analysis confirmed a significant lower expression of MSR-1 in LD-R-infected-KCs as compared to LD-S-infected-KCs (*Figure 7Di, ii*), thus explaining reduced staining for ox-LDL in response to LD-R-infection. Furthermore, increased expression of *atp6vod2* transcripts, implicated in LD-PV acidification and lipid utilization (*Pessoa et al., 2019*), was also noted to be slightly higher for LD-R-infected-PECs in the RNAseq (*Figure 6A*). However, western blot analysis showed a slightly increased expression of ATP6V0D2 in LD-R-infected-KCs as compared to LD-S-infected-KCs (*Figure 7E*), suggesting that it might also play a role in maintaining LD-PV acidity, facilitating lipid utilization and limiting ox-LDL uptake, as previously reported for *L. mexicana* infection (*Pessoa et al., 2019*). Finally, immunostaining of liver sections from 28 days LD-S and LD-R-infected mice (*Figure 7Fi*) confirmed a significant lower expression of MSR-1 in LD-R-infected liver with higher accumulation of neutral lipid droplets as compared to LD-S-infected murine liver (*Figure 7Fii, iii*), thus confirming LD-R specifically accumulates neutral lipids by restricting MSR-1 expression.

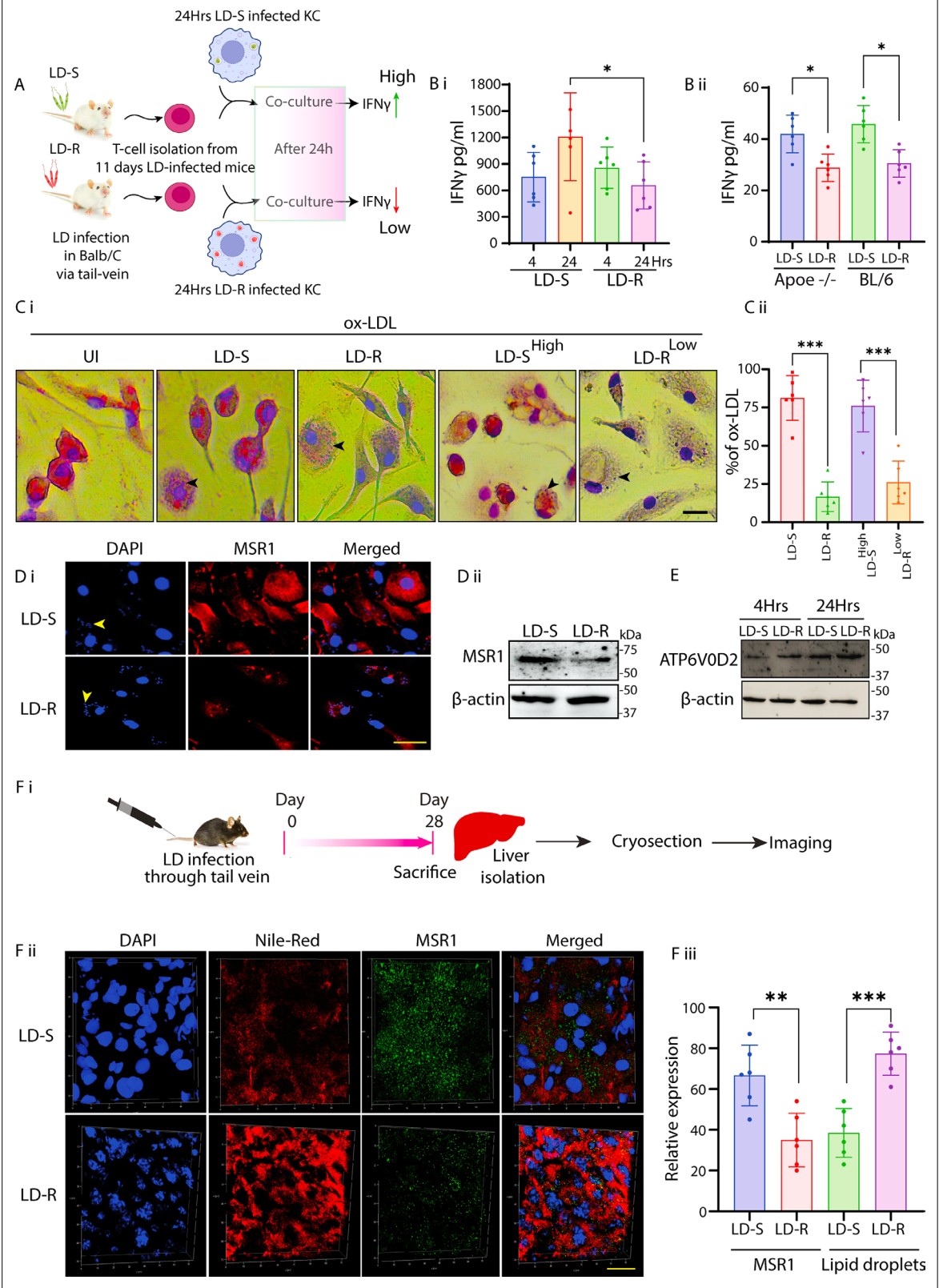

**Figure 7.** LD-R-infected-MΦs selectively excludes ox-LDL to suppress inflammatory response in host. (**A**) Scheme representing ex vivo co-culture experiment performed by isolating T-cells from LD-infected mice with in vitro LD-infected-KCs. (**B**) (i) IFN-γ level in the supernatant of LD-infected-KCs co-cultured with the T-cells isolated from spleen of Balb/c mice infected with LD-S or LD-R 28 days p.i. Data represented as mean ± SD from three independent experiments. (**B**) (ii) IFN-γ level in the supernatant of splenocytes cultures from LD-infected-BL/6 and *Apoe⁻/⁻* mice stimulated with SLA

*Figure 7 continued on next page*

*Figure 7 continued*

(soluble *Leishmania* antigen) specific for LD-S or LD-R. Data are presented as mean ± SD from six independent mice. (**C**) (i) ox-LDL incubated LD-infected-KCs was fixed and stained with Oil-red-O and hematoxylin. LD-S^High represents MOI (1:20) and LD-R^Low represents MOI (1:5). Scale bar: 10 μm. Black arrow represents LD nucleus. (**C**) (ii) Graph representing % of ox-LDL-positive MΦs under different experimental conditions as represented in (**C**) (i). (**D**) (i) Confocal image showing MSR-1 in LD-infected-KCs. Yellow arrow represents LD-amastigotes while N represents host nucleus. Scale bar: 20 μm. (**D**) (ii) Expression of MSR1 detected by western blot of LD-infected-KC with β-actin as loading control. (**E**) Expression of ATP6V0D2 in LD-infected-KCs by western blot with β-actin as loading control. (**F**) (i) Scheme representing the time line of in vivo experiment for IFA. (**F**) (ii) Expression of MSR1 and neutral lipid droplet accumulation assayed by confocal microscopy from the cryo-sectioned liver of 28 days LD-infected mice. Representative image showing lipid droplet marked in red and MSR1 in green, nucleus in blue. Scale bar: 40 μm. (**F**) (iii) Graphical representation of % positive lipid droplets accumulation and MSR-1 expression under different experimental condition as in F (**i**). Data collected from *N* = 6 mice. *** signifies p-value <0.0001, ** signifies p-value <0.001, * signifies p-value <0.05.

The online version of this article includes the following source data for figure 7:

**Source data 1.** Source data for western blots shown in *Figure 7Dii, E*.

**Source data 2.** Source for western blots shown in *Figure 7Dii, E*, with relevant bands labelled.

## Lipid droplets accumulation decreases Amphotericin-B responsiveness in LD-R amastigotes

Increased LDL uptake has been reported to cause high lipid droplets formation (*Lu and Gursky, 2013*; *Yano et al., 2022*), and lipid droplets formation has been reported to play critical role in intracellular proliferation of several pathogens, including LD-amastigotes (*Banerjee et al., 2022*; *Meester et al., 2011*; *Bosch et al., 2021*). As we have already observed significant accumulation of lipid droplets in LD-R-infected murine liver (*Figure 7*), we further investigated its role in response to LD-S and LD-R-infection. Lipid droplets staining of LD-S and LD-R-infected-KCs revealed significantly higher lipid droplets accumulation around LD-R amastigotes (*Figure 8—figure supplement 1A* and *Video 4*). For better visualization, we expanded (*Liffner et al., 2023*) EGFP-LD-R-infected-KCs and again performed lipid droplets staining on them which confirmed assimilation of neutral lipid into LD-R amastigotes (*Figure 8A*). This was further confirmed by performing transmission electron microscopy (TEM) in LD-R-infected-KC which showed lipid droplets accumulating around LD-R amastigotes (*Figure 8B*). Lipid accumulation has been previously reported to cause Amphotericin-B unresponsiveness among pathogens by hindering drug penetration (*Chang et al., 2015*). Interestingly, we noted that two LD-R-strains (BHU575 and BHU814) which we have included in this study also shows reduced responsiveness against Amp-B, a second-line-antileishmanial (*Ghosh et al., 2021*), and we validate this by determining their EC50 against Amp-B in LD-infected-KCs, which showed an EC50 of 0.39 and 0.31 μM for LD-R[1] and LD-R[2] amastigotes, respectively, as compared to EC50 of 0.03 μM for LD-S[1] and 0.032 μM LD-S[2]. We suspected that significant lipid droplets accumulation around LD-R amastigotes might be responsible for their reduced susceptibility toward Amp-B by limiting drug accessibility. To test this possibility, we performed independent infection with two LD-R strains in low and high lipid medium and access their EC50 against Amp-B. Low lipid medium results in significant drop in lipid droplets accumulation within LD-R-infected-KCs (*Figure 8—figure supplement 1B*) and also resulted in enhanced susceptibility for both the LD-R strains against Amp-B with EC50 reduced to 0.39–0.16 μM for LD-R[1] and 0.31–0.06 μM for LD-R[2] as compared to high lipid medium (*Figure 8Ci, ii*). Interestingly, infection with lab-generated Amphotericin-B-unresponsive LD promastigotes (LD-S^Amp-B-R) without primary unresponsiveness toward antimony did not result in any significant lipid droplets accumulation around LD-R amastigotes (*Figure 8—figure supplement 1A*) and neither showed any significant alteration in EC50 between low (0.09 μM) and high lipid medium (0.11 μM) (*Figure 8Ciii*). This observation clearly suggests that this high LDL-influx along with higher lipid droplet accumulation responsible for increased unresponsiveness of LD-R-isolates toward Amp-B is a specific attribute of LD isolates with primary unresponsiveness

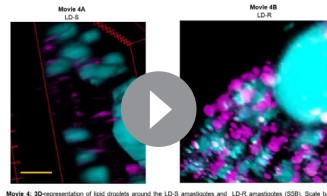

**Video 4.** 3D representation of lipid droplets around the LD-S and LD-R amastigotes (*Figure 8—figure supplement 1B*). Scale bar: 5 μm.

https://elifesciences.org/articles/102857/figures#video4

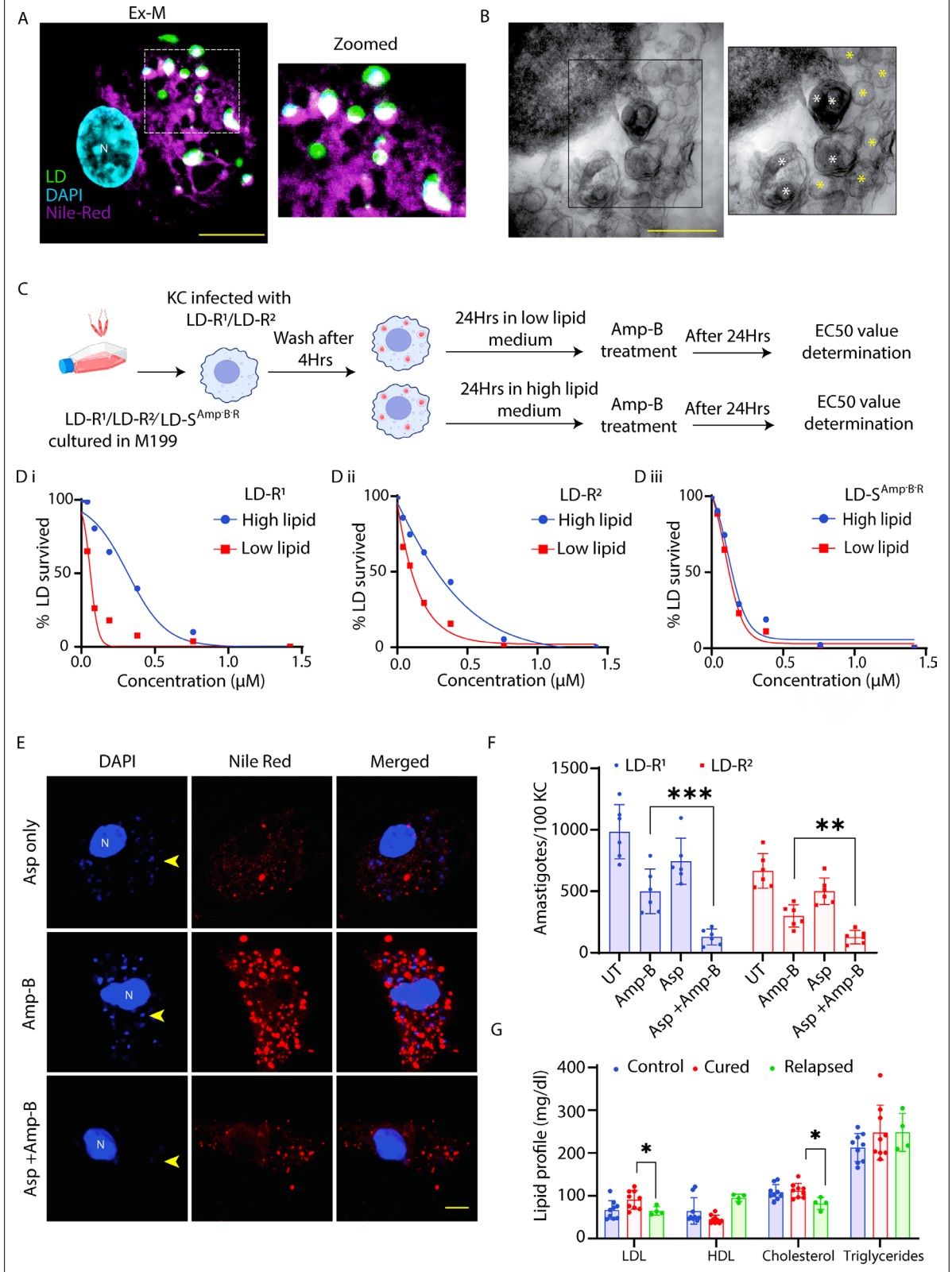

**Figure 8.** Lipid droplets accumulation around LD amastigotes is inversely related with susceptibility toward Amphotericin-B. (**A**) Ex-M image showing lipid droplet assimilation in LD-R amastigotes. White dotted region further cropped and shown in enlarged panel (Zoomed). Nile red (magenta), LD (green), and nucleus (cyan). White spots represent lipid droplets assimilated in LD-R and host nucleus is represented as N. Scale bar: 5 µm. (**B**) Transmission electron microscopy (TEM) images showing lipid droplets around the LD-R amastigotes. Black box further magnified and represented

*Figure 8 continued on next page*

*Figure 8 continued*

in right panel. White * representing LD-R amastigotes, yellow * showing lipid droplets. Scale bar 2 µm. (**C**) Schematic representation representing experimental scheme used to determine EC50 against Amphotericin-B under different experimental conditions. (**D**) (i) and (ii) Determination of EC50 value against Amphotericin-B in low and high lipid media by performing infection with of two independent LD-R strains. (**D**) (iii) Determination of EC50 value against Amphotericin-B in Kupffer cell (KC) infected with LD-S$^{Amp-B-R}$. Intracellular amastigotes were counted through Giemsa staining. (**E**) Confocal images showing lipid droplets accumulation around intracellular amastigotes with Aspirin (5 µM) or Amp-B (0.36 µM) or with a combination of Aspirin (5 µM) and Amp-B (0.11 µM)at 48 hr p.i. (**F**) Determination of amastigote load in KCs infected with two independent LD-R-isolates in different experimental conditions either untreated or treated with Aspirin (5 µM) or Amp-B (0.36 µM) or with a combination of Aspirin (5 µM) and Amp-B (0.11 µM) at 48 hr p.i. (**G**) Comparative lipid profile of visceral leishmaniasis (VL) patients (Cured and Relapsed) in response to Amp-B treatment along with healthy individuals from endemic region of Bihar, India. *** signifies p-value <0.0001, ** signifies p-value <0.001, * signifies p-value <0.05.

The online version of this article includes the following figure supplement(s) for figure 8:

**Figure supplement 1.** Aspirin treatment inhibits lipid droplets formation and increases responsiveness of LD-R amastigotes toward Amphotericin-B.

against antimony and may vary significantly from lab-generated Amp-B unresponsive strains. Previously, Aspirin (Asp), a lipid droplet blocker has been reported to block lipid droplet formation in *T.cruzi* and LD-infected-MΦs (*Banerjee et al., 2022*; *D'Avila et al., 2011*). We also observed a drastic drop in lipid droplets formation in LD-R-infected-KCs treated with Aspirin as compared untreated LD-R-infected-KCs (*Figure 8—figure supplement 1C*). Interestingly, while treatment with Aspirin alone did not significantly reduce the amastigote load in LD-R-infected-KCs, combining Aspirin with previously determined low dose of Amp-B (0.1 µM), completely restored Amp-B responsiveness against intracellular LD-R amastigotes (*Figure 8E, F*). This suggests that the enhanced uptake of LDL leading to lipid droplet accumulation might be a primarily responsible for increasing Amp-B unresponsiveness among clinical LD-R-isolates. Importantly, this observation is consistent with findings from human VL patients from the endemic region of Bihar, India, where those who relapsed after Amp-B treatment had significantly lower serum LDL and cholesterol levels compared to Amp-B treatment responders (*Figure 8G* and Table 2).

## Discussion

Our findings revolve around understanding how antimony-resistant *Leishmania donovani* (LD-R) has evolved as a persistent strain by adopting to a strategy of better utilization of host-derived lipids to sustain their own aggressive intracellular proliferation. Intracellular pathogens often alter the host's immune-metabolic state for better survival (*Thakur et al., 2019*; *Kreimendahl and Pernas, 2024*). Our observations pointed out that high organ parasite load linked with LD-R-infection does not rely only on initial infectivity as commonly believed to date (*Vanaerschot et al., 2011*), rather is indicative of a selective higher intracellular proliferation in host. We observed LD-R amastigotes scrutinize and access specific type of host lipoproteins to support their aggressive proliferation as seen in clinical infection (*Figure 1*, *Figure 1—figure supplement 1*; *Lal et al., 2007*). Previously increased amastigote load in VL patients has been inversely linked with their serum cholesterol level (*Ghosh et al., 2011*), and we also observed that mice infected with LD-R-isolates had significantly lower HDL, LDL, and cholesterol compared to LD-S-infected mice (*Figure 1*).

LD amastigotes are lipid auxtrophs (*Goad et al., 1984*; *Bouazizi-Ben Messaoud et al., 2017*), and this severe dyslipidemia in response to LD-R-infection can be logically interpreted in terms of LD-R's increase need to support their aggravated intracellular proliferation in the infected host. In liver-resident macrophages (KCs), primary sites for LD infection and host cholesterol synthesis, we observed significantly higher number of LD-R amastigotes and PVs formation, compared to LD-S-infection, indicating LD-R's better capacity to utilize host cholesterol (*Figure 1*, *Figure 1—figure supplement 1*). Also, in hyperlipidemic *Apoe*$^{-/-}$ mice, LD-R-infection led to organ parasite overload compared to LD-S-infection, confirming LD-R's increased dependence on extracellular lipid sources to sustain its aggressive proliferation (*Figure 2*). Thus, next we focused on investigating LD-R's ability to differentially regulate host's immune-metabolic status. Previous reports suggested that LD with altered drug responsiveness can modulate their host differentially resulting in different disease outcome (*Mukherjee et al., 2013*; *Díaz Varela et al., 2024*). Our results confirmed that while both LD-R and LD-S promastigotes initially quench cholesterol from the host plasma membrane similarly (*Figure 3—figure supplement 1*), LD-R amastigotes accumulate significantly more cholesterol and fatty acids in

their PV by specifically promoting increased endocytosis of LDL as the infection progresses (*Figure 3*, *Figure 3—figure supplement 2*). This possibly also explains the similar initial attachment and infectivity of LD-S and LD-R promastigotes with the host MΦs again indicating toward their ability to infect their host equally (*Figure 1*, *Figure 1—figure supplement 1*, *Video 1*).

Interestingly, this LD-R-induced LDL-endocytosis is independent of the host's LDL receptor and relies on fluid-phase endocytosis, involving significant actin cytoskeleton reorientation in LD-R-infected-MΦs (*Figure 4*, *Figure 4—figure supplement 1*). It is worth mentioning that LD-R-infection has been reported to induce high levels of IL-10 (*Mukherjee et al., 2013*), which is linked with enhanced fluid-phase endocytosis (*Lucero et al., 2020*).

Importantly, we also noted an increased convergence of endocytosed LDL-vesicles with lysosomal associated marker protein (LAMP-1), along with higher LAMP-1 expression around LD-R amastigotes as compared to LD-S ones (*Figure 5*). Lysosomal acid hydrolases metabolize LDL into bioavailable cholesterol, and higher fusion of lysosomal vesicles with LD-R-PV explains the significant Filipin-cholesterol staining around LD-R amastigotes (*Figure 5*). These findings not only confirm that endocytosed-LDL can be better utilized by LD-R amastigotes but probably also indicates toward a rapid maturation of LD-R-PV due to increased LDL-vesicle fusion (*Sood et al., 2024*; *Pessoa et al., 2019*).

Although available RNAseq analysis (*Figure 6*) did not support this increased expression of *lamp-1* in the transcript level, it did reflect a notable upregulation of vesicular fusion protein (VSP) *vamp8* and *stx1a* in response to LD-R-infection. How, LD infection can regulate LAMP-1 expression, and the role of VSPs in LDL-vesicle fusion with LD-R-PV is worthy of further investigation. It is possible and has been earlier reported that LD infection can regulate host proteins expression through post transcriptional and post translational modifications (*Geraci et al., 2015*; *Tiwari et al., 2017*; *Moradimotlagh et al., 2023*). It is tempting to speculate that LD-R amastigote might be promoting an increased lysosomal biogenesis through any such mechanism to increase supply of bioavailable cholesterol through action of lysosomal acid hydrolases on LDL. Additionally, RNAseq analysis also showed suppressed expression of *Npc1* in LD-R-infected-MΦs. NPC-1 is crucial for mobilizing vesicular cholesterol to the cytoplasm and plasma membrane (*Urano et al., 2008*), and its suppressed expression correlates with higher cholesterol availability for aggressive LD-R proliferation (*Figure 6*).

Besides LDL endocytosis, de novo cholesterol synthesis could also serve as a lipid source for intracellular LD amastigotes (*Boodhoo et al., 2020*). Interestingly, LD-R-infected macrophages exhibited suppressed expression of HMGCR and SREBP2, the key regulators of de novo cholesterol biosynthesis. Additionally, SREBP2 showed minimal nuclear translocation in LD-R-infected macrophages compared to LD-S-infected cells, (*Figure 6*), consistent with reports of a shutdown in de novo cholesterol synthesis in MΦs with high endocytosed LDL accumulation (*Prakash and Rai, 2022*). Notably, SREBP2 regulates NPC-1 expression (*Garver et al., 2008*) and has been shown to get activated in response to LD-S-infection (*Prakash and Rai, 2022*). Further RNA sequencing data also revealed a significant downregulation of *hmgcs* (3-hydroxy-3-methylglutaryl-CoA synthase) in LD-R-infected PECs as compared to LD-S-infection. Downregulation of HMGCS which catalyzes the condensation of acetyl-CoA with acetoacetyl-CoA to form 3-hydroxy-3-methylglutaryl-CoA (HMG-CoA), which serves as an intermediate in both cholesterol biosynthesis and ketogenesis further supports our observation that LD-R-infected PECs preferentially rely on endocytosed-LDL-derived cholesterol rather than de novo synthesized cholesterol to support their metabolic needs.

Surprisingly, LD-R-infected-MΦs showed minimal inflammatory response despite high lipid accumulation, due to their ability to successfully exclude uptake of ox-LDL. Low expression of MSR-1, a scavenger receptor of ox-LDL, in the liver of LD-R-infected mice explained this observation, leading to the accumulation of neutral lipid droplets around LD-R amastigotes (*Figure 7*, *Figure 8—figure supplement 1*, *Video 4*). Expansion microscopy (*Liffner et al., 2023*) along with TEM confirmed significant assimilation of these neutral lipids into LD-R amastigotes (*Figure 8*). Several intracellular pathogens like *Mycobacterium*, *Toxoplasma*, HCV have been reported to transfer stored triacylglycerols and sterol esters from lipid droplets to support their proliferation (*Sonda and Hehl, 2006*; *Kim and Shin, 2023*; *Awadh, 2023*). While the understanding of lipid uptake and metabolizing genes in LD is limited, a recent genome-wide CRISPR screen in *L. mexicana* (*LeishGEM, 2024*), have identified such potential candidates 'essential' for parasite survival. Close proximity lipid droplets with LD-R amastigotes makes it tempting to speculate existence of lipid droplets contact sites (*Sood et al., 2024*), or possible copy number variations in these lipid metabolizing genes (*Downing et al., 2011*),

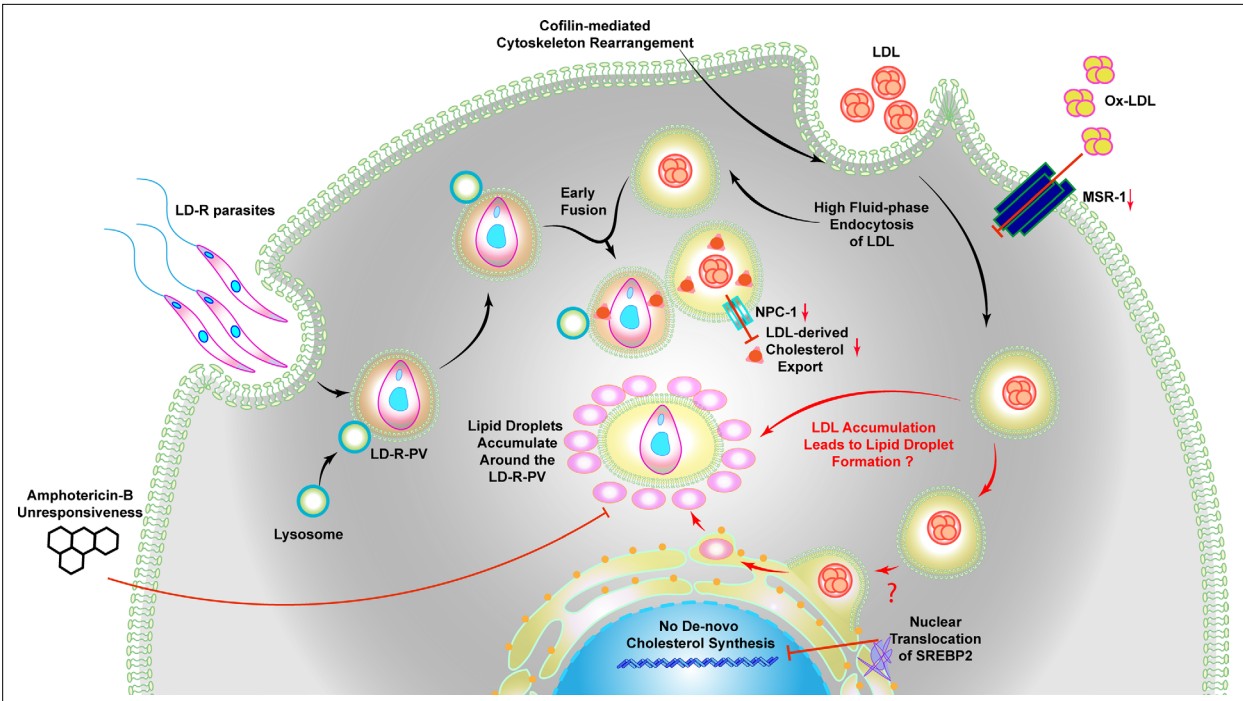

**Figure 9.** Scrutinized uptake of host lipids results in breakdown of Amphotericin-B responsiveness in clinical LD isolates with primary resistance to antimony. LD-R reside within membrane-bound parasitophorous vacuoles (PVs) and actively modulate host actin cytoskeleton dynamics through cofilin-mediated depolymerization, inducing high fluid-phase endocytosis of low-density lipoprotein (LDL). The internalized LDL-containing vesicles subsequently fuse with LD-R-associated PVs, facilitating increased lipid acquisition essential for their aggressive intracellular proliferation. In these fused membranes, the vesicular cholesterol export protein NPC-1 is downregulated, effectively sequestering LDL-derived cholesterol within the PV and preventing its release into the host cytoplasm. Concurrently, LDL-derived cholesterol esters either directly or getting processed in the endoplasmic reticulum (ER) can lead to the formation of lipid droplets (as presented by ?, since it needs further investigation). Lipid droplets then accumulate in and around the LD-R-PV. This lipid droplet accumulation is a key factor in rendering LD-R-infections unresponsive to Amphotericin-B. Additionally, LD-R regulate inflammatory responses linked to oxidized lipid (ox-LDL) accumulation by downregulating macrophage scavenger receptor 1 (MSR-1). The elevated LDL-influx also leads to a shutdown of de novo cholesterol biosynthesis by restricting the nuclear translocation of sterol regulatory element-binding protein 2 (SREBP2).

providing survival advantages to LD-R. This speculation is further supported by the fact that although clinical LD-R-isolates with neutral lipid droplet accumulation in infected MΦs displayed increased unresponsiveness to Amp-B, lab-generated Amp-B unresponsive promastigotes failed to do so in infected MΦs (*Figure 8*). Fact that LD-R amastigotes regained their responsiveness toward Amp-B in low lipid medium or in presence of Aspirin (*Banerjee et al., 2022*), further confirms that eliminating lipid droplets around LD-R amastigotes might be a weapon of choice to combat treatment failure against Amp-B (*Purkait et al., 2012*; *Srivastava, 2011*; *Patole et al., 2014*). Our results thus suggest that natural clinical LD-R-isolates have developed an inherent adaptability to gradually develop cross-resistance toward Amp-B, probably as an additional consequence of their ability to accumulate increased host-derived lipids as a rapidly proliferating intracellular amastigote (represented graphically in *Figure 9*). Our work explains increasing Amp-B treatment failure among VL patients (*Purkait et al., 2012*; *Srivastava, 2011*; *Patole et al., 2014*) and correlates with low serum LDL and cholesterol levels in patients relapsing after Amp-B treatment (*Figure 9*, Table 2).

## Methods
### Reagents
Laurdan (Merck), photoRED-LDL (Thermo Fisher Scientific), NBD Cholesterol (Sigma), Nile red (Sigma-Aldrich), Oil-red-O (Thermo Fisher Scientific), Filipin (Sigma-Aldrich), and Collagenase (Gibco) were purchased. Tissue culture plasticware were purchased from NUNC (Roskilde, Denmark). Invitrogen Amplex-Red assay kit was purchased from Thermo Fisher Scientific. Primary antibodies required

for the experiments were anti-ATP6V0D2 (ABclonal), anti-MSR1 (ABclonal), anti-LAMP1 (Thermo Fisher Scientific), anti-SREBP2 (Thermo Fisher Scientific), anti-NPC1 (Santa Cruz Biotechnology), anti-HMGCR (Santa Cruz Biotechnology), anti-LDLr (Santa Cruz Biotechnology), anti-KMP-11 (Invitrogen), goat anti-mouse IgG Alexa Fluor 488 (Thermo Fisher Scientific), goat anti-mouse IgG Alexa Fluor 594 (Thermo Fisher Scientific), goat anti-Rabbit IgG-HRP (Thermo Fisher Scientific), goat anti-Mouse IgG-HRP (Thermo Fisher Scientific), β actin (Cell Signalling Technology), CD11b (Cell Signalling Technology), Cofflin (Cell Signalling Technology), and phospho-Cofflin (Cell Signalling Technology). Rhodamine Phalloidin reagent was purchased from Thermo Fisher Scientific and ELISA Kits for IFN-γ were purchased from R&D Systems, Aspirin generic composition, SAG (Albert David).

## Animals

BALB/c, C57BL/6, and $Apoe^{-/-}$ mice were maintained and bred under pathogen-free conditions with food supplements and filtered water. All experiments were conducted in accordance to Institutional Animal Ethics Committee (IAEC), IIT Kharagpur guidelines India (IAEC/BI/1 55/2021) and approved by National Regulatory Guidelines issued by Committee for the Purpose of Supervision of Experiments on Animals, Ministry of Environment and Forest, Government of India.

## Ethical statement

In this study, all procedures involving human participants were conducted in accordance with the ethical standards of the 1964 Helsinki Declaration and approval of the 'Institutional Human Ethical Committee' of ICMR-Rajendra Memorial Research Institute of Medical Sciences, Patna, India (Approval No. RMRI/EC/20/2020). Following information about the potential risks, benefits, and the investigational nature of the study consent from all study participants was obtained in the informed consent form.

## Parasite cultures and maintenance

*L. donovani* (LD) isolates used in this study are natural clinical isolates obtained from the patients MHOM/IN/09/BHU569/0, MHOM/IN/2009/BHU575/0, MHOM/IN/10/BHU814/1, MHOM/IN/83/AG83, MHOM/IN/09/BHU777/0, and MHOM/IN/80/DD8 (*Mukhopadhyay et al., 2011*; *Mukherjee et al., 2013*), and were maintained in BALB/c mice as per institute biosafety guidelines (BT/BS/17/27/97-PID). Susceptibility of each of these LD isolates against antimony was checked and confirmed against previously reported EC50 value for antimony before they were included in the study (this can be reported in a common table, *Table 1*). Lab-generated Amp-B resistant strain (DD8-AmpB) strain was kind from Dr. Arun Kumar Haldar (CSIR-CDRI). Amastigotes were isolated by macerating the spleen of 21 days infected mice under sterile conditions, and subsequently transformed into promastigotes by maintaining in M199 (Gibco) supplemented with 5% (vol/vol) heat-inactivated fetal bovine serum (FBS) (Gibco) and 1% (vol/vol) penicillin–streptomycin solution (Gibco) under drug selection (*Table 2*).

**Table 1.** EC50 against SAG (antimonial) for the LD strains used in this study.

| Strain name | EC50 value for SAG (µg/ml) in PEC | EC50 value for SAG (µg/ml) in KC |
|---|---|---|
| MHOM/IN/2009/BHU575/0 | 22.73 ± 3.7 | 23.89 ± 2.1 |
| MHOM/IN/10/BHU814/1 | 30.3 ± 7.2 | 28.5 ± 5.3 |
| MHOM/IN/83/AG83 | 1.52 ± 1.2 | 3.3 ± 2.2 |
| MHOM/IN/80/DD8 | 2.1 ± 1.2 | 3.2 ± 1.9 |
| DD8-AmpB | 1.8 ± 2.1 | 2.2 ± 1.3 |
| BHU575-EGFP | 21.45 ± 3.2 | 22.81 ± 3.1 |
| AG83-EGFP | 2.1 ± 1.4 | 2.4 ± 3.1 |

Magenda shaded: drug unresponsive strains; green shaded: drug responsive strains.

**Table 2.** Lipid profile of visceral leishmaniasis (VL) patients.

| Serum code | Age | Sex | Time of blood collection | Infection status | Treatment received | Patient status | LDL (rg/d) | HDL (rrg/dl) | Cholesterol (mg/dl) | Tridycerides (mg/dl) |
|---|---|---|---|---|---|---|---|---|---|---|
| RMRI N1 | 45 | F | Before | Control population from endemic zone | NA | NA | 45.3501585 | 46.5365931 | 101.6949153 | 177.258567 |
| RMRI N6 | 70 | F | Before | Control population from endemic zone | NA | NA | 110.855478 | 50.3762712 | 111.8644068 | 206.5420561 |
| RMRI N7 | 45 | F | Before | Control population from endemic zone | NA | NA | 75.5620787 | 41.8711864 | 84.40677966 | 165.4205607 |
| RMRI N8 | 19 | M | Before | Control population from endemic zone | NA | NA | 83.946038 | 62.9533123 | 100.3389831 | 224.9221184 |
| RMRI N11 | 37 | M | Before | Control population from endemic zone | NA | NA | 72.4549921 | 120.914196 | 134.2372881 | 244.5482866 |
| RMRI N12 | 60 | M | Before | Control population from endemic zone | NA | NA | 52.6307448 | 113.851735 | 92.54237288 | 180.6853583 |
| RMRI N13 | 30 | M | Before | Control population from endemic zone | NA | NA | 66.7533281 | 48.3413249 | 138.3050847 | 236.4485981 |
| RMRI N14 | 22 | M | Before | Control population from endemic zone | NA | NA | 47.8939778 | 49.6807571 | 100 | 260.7476636 |
| RMRI N15 | 37 | F | Before | Control population from endemic zone | NA | NA | 46.4904913 | 45.9059937 | 108.8135593 | 219.6261682 |
| RMRI CI | 20 | F | Before | Confirmed VL, hospitalization | Ambisome Single dose | Cured | 110.389045 | 44.2700565 | 112.5423729 | 200.623053 |
| RMRI C2 | 62 | F | Before | Confirmed VL, hospitalization | Ambisome Single dose | Cured | 121.738904 | 64.5514124 | 145.4237288 | 3018691589 |
| RMRI C3 | 15 | M | Before | Confirmed VL, hospitalization | Ambisome Single dose | Cured | 76.1839888 | 36.7463277 | 95.59322034 | 185.6697819 |
| RMRI C4 | 31 | M | Before | Confirmed VL, hospitalization | Ambisome Single dose | Cured | 72.141573 | 41.1079096 | 116.9491525 | 241.7445483 |
| RMRI C5 | 52 | M | Before | Confirmed VL, hospitalization | Ambisome Single dose | Cured | 98.1063202 | 48.1954802 | 117.2881356 | 282.2429907 |

*Table 2 continued on next page*

*Table 2 continued*

| Serum code | Age | Sex | Time of blood collection | Infection status | Treatment received | Patient status | LDL (rg/d) | HDL (rrg/dl) | Cholesterol (mg/dl) | Tridycerides (mg/dl) |
|---|---|---|---|---|---|---|---|---|---|---|
| RMRI C6 | 18 | F | Before | Confirmed VL, hospitalization | Ambisome Single dose | Cured | 68.8765449 | 36.7463277 | 96.61016949 | 200.623053 |
| RMRI C7 | 45 | F | Before | Confirmed VL, hospitalization | Ambisome Single dose | Cured | 110.544522 | 55.6101695 | 124.4067797 | 381.9314642 |
| RMRI C8 | 32 | F | Before | Confirmed VL, hospitalization | Ambisome Single dose | Cured | 94.8412921 | 43.0706215 | 106.1016949 | 232.3987539 |
| RMRI C18 | 66 | M | Before | Confirmed VL, hospitalization | Ambisome Single dose | Cured | 61.4136236 | 35.9830508 | 91.86440678 | 203.4267913 |
| RMRI R14 | 8 | M | Before | Confirmed VL, hospitalization | Ambisome Single dose | Relapse | 79.2935393 | 83.6333333 | 96.94915254 | 262.9283489 |
| RMRI R16 | 49 | M | Before | Confirmed VL, hospitalization | Ambisome Single dose | Relapse | 56.9047753 | 102.279096 | 63.05084746 | 209.0342679 |
| RMRI R17 | 18 | M | Before | Confirmed VL, hospitalization | Ambisome Single dose | Relapse | 59.7033708 | 101.079661 | 83.38983051 | 217.1339564 |
| VRMRI R18 | 59 | M | Before | Confirmed VL, hospitalization | Ambisome Single dose | Relapse | 63.4348315 | 94.319209 | 84.40677966 | 304.6728972 |

Green shaded: healthy; yellow shaded: cured; magenta shaded: relapsed patients.

## Cell culture and infection

Peritoneal exudate cells (PECs) were harvested from BALB/c mice as mentioned in already published protocol, with slight modifications (*Ray and Dittel, 2010*). After 48 hr of 4% starch intraperitoneal injection, PECs were isolated and plated on 35 mm tissue culture Petri dishes, sterile 22 mm square coverslips in 35 mm Petri dishes or 24-well plates at a density of $1 \times 10^6$, $1.5 \times 10^5$, or $1.5 \times 10^5$, respectively, all in RPMI 1640 medium (Gibco) supplemented with 10% (vol/vol) heat-inactivated FBS, 1% (vol/vol) penicillin–streptomycin solution. The cells were allowed to adhere for 48 hr at 37°C in 5% $CO_2$ before the infection process.

## In vivo infection

For in vivo infections, healthy female C57BL/6 mice or *Apoe*$^{-/-}$ mice (4–6 weeks old and 20–25 g each) were injected via tail vein route with $1 \times 10^7$ GFP-expressing *L. donovani* stationary phase metacyclic promastigotes (suspended in 100 µl 1× phosphate-buffered saline [PBS]) as reported previously (*Khanra et al., 2020*). After infection mice were kept in atherogenic diet till sacrifice. Each group contained five mice that were sacrificed 11 days' post-infection (p.i). Splenocytes were isolated and GFP-positive population was measured through flow cytometry. Each experiment was performed in triplicate.

## Isolation of KCs

KCs were harvested from mice livers following a previously published protocol, with slight modifications (*Li et al., 2014*). Wild-type BALB/c mice were anesthetized, and warm PBS containing 0.05% collagenase IV was used for liver perfusion. The livers were swiftly excised and minced into small fragments on an ice bath. Tissue digestion was performed using a 0.1% Type IV Collagenase solution in RPMI 1640 for 30 min at 37°C with intermittent shaking for every 10 min. Following digestion, KCs were separated from hepatocytes and other cellular components through sequential centrifugation steps and plated on well plates, as required, for 4 hr in RPMI 1640 media supplemented with 10% (vol/vol) penicillin– streptomycin and 10% (vol/vol) FBS. After 24 hr of resting at 37°C in 5% $CO_2$, the infection studies commenced. To assess the purity of the isolated KCs by flow cytometry, KCs were plated on a 6-well plate. After 4 hr, KCs were scraped out, centrifuged at $300 \times g$ for 5 min and stained with monoclonal anti-F4/80 PE conjugated antibody (BD Biosciences). The KCs were then washed and fixed with 2% paraformaldehyde (PFA) for 5 min, analyzed using a flow cytometer (ARIA FACS DIVA), and further processed using FLOW JO software for data analysis. Purified KCs were infected with stationary phase sorted metacyclic LD promastigotes as mentioned previously. PECs and KCs are collectively designated as MΦs (macrophages).

## Sorting of metacyclic LD and infection in MΦs

Metacyclic parasites were sorted through Beckman Coulter CytoFLEX cell sorter, as per the manufacturer's protocol. The LD promastigotes were washed and resuspended in PBS before being subjected to light scatter analysis using flow cytometry. Sorted population were checked with forward scatter (FSC) versus side scatter (SSC) and dot plot was generated to represent the acquisition of 10,000 events using a FACS Aria II system. Required number of sorted metacyclic LD was used to perform subsequent MΦ and mice infections. MΦs were infected with stationary-phase flow sorted metacyclic LD promastigotes at an MOI 1:10 ratio (in some cases 1:5 and 1:20) for 4 hr which was considered as the 0th point of the infection. MΦs was then washed to remove any extracellular loosely attached parasites and incubated further as per experimental requirement.

## Intracellular amastigote load determination

To enumerate intracellular amastigote load, resting stage PECs and KCs on glass coverslips were infected with sorted metacyclic LD promastigotes of various strains at a ratio of 1:10, as described in previous protocols (*Mukherjee et al., 2013*; *Mukherjee et al., 2020*). Following a 4-hr incubation, the PECs or KCs were washed and subsequently cultured for specific durations. Coverslips were retrieved after specific time points, rinsed with PBS, fixed using methanol, and stained with Giemsa. These prepared coverslips were then affixed onto glass slides and examined utilizing a light microscope (Olympus). Parasite quantification was conducted by counting amastigotes per 100 macrophages. Image processing related to *Figure 1A*, *Figure 1—figure supplement 1D* was performed using Fiji

by separating the spectral overlap between the light and dark stained region to Giemsa stained MΦs. Next, for better visualization, pseudo colors were assigned to represent nucleus (white) and cytoplasm (red) with a black background. Color separated images were then merged showing LD within PV.

## Raman spectroscopy of LD-infected-KCs

KCs were plated on Raman-grade stainless steel plate and kept for 24 hr in a humidified environment at 37°C with 5% $CO_2$, followed by infection with LD as already described. At 24 hr p.i., the KCs were washed using PBS, fixed with 2% PFA and proceeded to perform Raman spectroscopy (Witec, alpha300R). The Raman grade stainless steel plates were positioned on the microscope stage and focused with a 100× air objective and measurements were performed using a 532-nm laser source producing a laser spot of approximately 0.3 µm with 20 mW laser power. Spectral measurements were obtained through the Project FIVE interface (Witec, Germany) after setting the grating to 600 gratings/mm. The spectral resolution was set to 2.87 cm⁻¹. The Raman spectra were taken with integration time set to 5 s, and 10 accumulations were selected after confirming its effectiveness on the KCs being studied. Baseline correction and other spectral denoising procedures were carried out using MATLAB 2017b (MathWorks), and the processed spectra were analyzed as described preciously (*Feuerer et al., 2021*; *Tiwari et al., 2020*). For confocal Raman spectroscopy, spectral data were acquired from individual cells at ×1000 magnification using a 100 × 100 µm scanning area, following previously established specifications. After spectral acquisition, distinct Raman shifts corresponding to specific biomolecular signatures were extracted for further analysis. These included: cholesterol (535–545 cm⁻¹), nuclear components (780–790 cm⁻¹), lipid structures (1262–1272 cm⁻¹), and fatty acids (1436–1446 cm⁻¹). Following spectral extraction, pseudo-color mapping was applied to highlight the spatial distribution of each biomolecular component within the cell. These processed spectral images are presented in *Figure 3D1*, where the first four panels illustrate the individual biomolecular distributions. A merged composite image was then generated to visualize the co-localization of these biomolecules within the cellular microenvironment, with the final panel specifically representing the spatial distribution of key biomolecules.

## Quantification of membrane cholesterol with Amplex Red assay kit

The KC membrane preparation was conducted following the method described previously (*Ghosh et al., 2012*), with slight modifications. KCs were ruptured by repeated freezing and thawing cycles followed by probe sonicating for four cycles with 30 s burst and 30 s gap. The resulting homogenate was then centrifuged at 900 × *g* for 10 mins at 4°C. The supernatant was filtered through a nylon mesh (100 µm), while the pellet was discarded. The filtered supernatant was centrifuged at 100,000 × *g* for 1 hr at 4°C, and resulting pellet, representing native membrane, was suspended in a buffer containing 50 mM Tris (pH 7.4), 1 mM EDTA, 0.24 mM PMSF, and 10 mM iodoacetamide. The protein content of the membrane was quantified using Bradford reagent (Bio-Rad), following the manufacturer's protocol. The total cholesterol content was determined using an Amplex Red reagent kit (*Amundson and Zhou, 1999*), with the results expressed in moles of cholesterol per gram of protein.

## PV isolation and cholesterol measurement

PVs were isolated using a previously outlined protocol with slight modifications (*Real, 2020*). $10^7$ KCs were seeded in a 100-mm plate and allowed to adhere for 24 hr. Following this infection was performed with LD for 24 hr, the infected KCs were then harvested by gentle scraping and lysed through five successive passages through an insulin needle to ensure membrane disruption while preserving organelle integrity. The lysate was centrifuged at 200 × *g* for 10 min at 4°C to remove intact cells and large debris. The resulting supernatant was carefully collected and subjected to a discontinuous sucrose density gradient (60%, 40%, and 20%). The gradient was centrifuged at 700 × *g* for 25 min at 4°C to facilitate organelle separation. The interphase between the 40% and 60% sucrose layers, enriched with PVs, was carefully collected and subjected to a final centrifugation step at 12,000 × *g* for 25 min at 4°C. The supernatant was discarded, and the resulting pellet was enriched for purified PVs, suitable for downstream biochemical and molecular analyses. Cholesterol and protein contents in PV were determined by an Amplex Red assay kit and Bradford assay, respectively. Resulting data were represented as micrograms of cholesterol per microgram of protein.

## GC–MS analysis of LD-S and LD-R-PV

Following a 24-hr infection period, KCs were harvested, washed with PBS, and pelleted. Subsequent to this, PV isolation was carried out using the previously described protocol (*Pessoa et al., 2019*). After PV isolation Bradford assay was carried out for normalizing the protein concentration. The resulting equal volume of PV pellet was suspended in 20 ml of dichloromethane: methanol (2:1, vol/vol) and incubated at 4°C for 24 hr. After centrifugation (11,000 × *g*, 1 hr, 4°C), the supernatant was checked through TLC and subsequently evaporated under vacuum. The residue and pellet were saponified with 30% potassium hydroxide in methanol at 80°C for 2 hr. Sterols were extracted with n-hexane, evaporated, and dissolved in dichloromethane. A portion of the clear yellow sterol solution was treated with *N,O*-bis(trimethylsilyl)trifluoroacetamide and heated at 80°C for 1 hr to form trimethylsilyl ethers. GC/MS analysis was performed using a Varian model 3400 chromatograph equipped with DB5 columns (methyl-phenylsiloxane ratio, 95/5; dimensions, 30 m by 0.25 mm). Helium was used as the gas carrier (1 ml/min). The column temperature was maintained at 270°C, with the injector and detector set at 300°C. A linear gradient from 150 to 180°C at 10 °C/min was used for methyl esters, with MS conditions set at 280°C, 70 eV, and 2.2 kV (*Alpizar-Sosa et al., 2022*).

## Cytokine measurement

Supernatants from KCs infected with stationary-phase LD co-culture with T cells, isolated from the mice infected with LD. For detection of IFN-γ from *Apoe*$^{-/-}$, splenocytes isolated from LD infected mice and cultured with SLA. IFN-γ levels in the supernatants were measured using a sandwich ELISA Kit (R&D Systems) as per the manufacturer's protocol. Detection limit for these kits was 8 pg/ml for IFN-γ. Cytokine levels were determined by measuring the OD at 450 nm using a MultiSkan FC microplate photometer (Thermo Fisher Scientific). Data were demonstrated as mean ± SD of all five individual experiments.

## Western blot analysis

Lysates of LD-infected PECs and KCs were prepared, and western blotting was performed for different proteins with endogenous control β-actin (Cell Signaling Technology). Blots were probed with specific antibodies. Binding of secondary HRP-labeled goat anti-rabbit or goat anti-mouse Abs was analyzed using SuperSignalR West Pico or West Dura Chemiluminescent substrate (Pierce).

## Confocal and structural illumination microscopy

PECs and KCs were cultured on 13 mm diameter glass coverslips at a density of $2 \times 10^5$, and infected with LD as mentioned previously. After varying time intervals, infected PECs or KCs were rinsed with PBS, fixed with 2% PFA, and permeabilized using 0.2% Triton/PBS for 20 min on an orbital shaker. Non-specific binding was blocked with 2% BSA in 0.2% Triton/PBS for another 20 min on an orbital shaker. The PECs and KCs were then incubated with primary antibodies for 1 hr, followed by washing and staining with IgG Alexa Fluor 594 or IgG Alexa Fluor 488, as required. Mounting was performed using Fluromount-G-DAPI (Thermo Scientific). Images were captured using Olympus confocal microscopy (FV3000) and super-resolution microscopy ZEISS (SIM) with a 63× objective. 3D image visualization processed through Fiji 3D viewer and 3D project plugin. Live cell microscopy was performed using ZEISS phase contrast microscope and ZEISS Apotome. MΦs were plated on confocal dish and incubated with photoRED-LDL for live cell imaging of LDL uptake (*Video 3*).

## Image processing and analysis

Image processing and analysis were conducted using Fiji (ImageJ). For optimal visualization, Giemsa-stained macrophages (MΦs) were represented in gray scale to enhance contrast and structural clarity. To improve the distinction of different fluorescent signals, pseudo-colors were assigned to fluorescence images, ensuring better differentiation between various cellular components. For colocalization analysis (*Figures 3, 5, and 6*, *Figure 3—figure supplement 1*), we utilized the RGB profile plot plugin in ImageJ, which allows for the precise assessment of signal overlap by generating fluorescence intensity profiles across selected regions of interest. This approach provided quantitative insights into the spatial relationship between labeled molecules within infected cells. Additionally, for analysing the distribution of cofilin in *Figure 4*, the ImageJ surface plot plugin was employed. This tool enabled

three-dimensional visualization of fluorescence intensity variations, facilitating a more detailed examination of cofilin localization and its potential reorganization in response to infection.

### Lipid droplet staining

LD-infected-KCs were first fixed using 2% PFA and subsequently permeabilized with 0.25% Triton X-100. Coverslips were then treated with Nile red (Sigma, diluted 1:100) in PBS for at least 1 hr to stain lipids, followed by DNA staining with DAPI. The coverslips were mounted onto glass slides using Fluoromount-G and visualized with confocal microscope (LEICA STELLARIS 5). 3D stacks were acquired to visualize distribution of lipid droplets. Images were processed and analyzed, and stacks were merged to reconstruct 3D images using Fiji.

### Ultrastructure expansion microscopy

Ultrastructure expansion microscopy was conducted on LD-infected-MΦs following established experimental protocols (*Liffner et al., 2023*). Protein crosslinking was initiated by immersing the coverslip in a solution containing 1.4% formaldehyde, 2% acrylamide, and PBS for 5 hr at 37°C. Monomer solution (19% sodium acrylate, 10% acrylamide, and 0.1% *N,N'*-methylenbisacrylamide in PBS) was mixed with 10% TEMED and 10% APS solution on ice and added to 6-well pates. Gelation was performed on ice, with cell-plated coverslip placed face down on the gelation solution in a humid chamber for 5 min at 37°C for 1 hr. The gel and coverslip were together transferred to a 6-well plate filled with denaturation buffer (200 mM SDS, 200 mM NaCl, and 50 mM Tris pH 9.0) and incubated face up for 15 min at room temperature under agitation, followed by incubation at 95°C for 1.5 hr. The first expansion phase involved incubating the gel three times in ddH$_2$O for 30 min each. Previously mentioned IFA steps were adopted for antibody staining. Images were captured using an Olympus confocal microscope (FV3000) equipped with a 63× 1.4 NA oil objective and Small Volume Computational Clearing mode was used to obtain deconvolved images. 3D stacks were acquired and analyzed using Fiji software.

### Transmission electron microscopy

For TEM, PECs infected with the LD-R strain were fixed after 24 hr of infection using 2.5% glutaraldehyde, an electron microscopy (EM) grade fixative. The cells were then processed according to previously established protocols (*D'Avila et al., 2011*). Following this, the samples were examined using a JEM-F200 transmission electron microscope, allowing for lipid droplet imaging and detailed visualization of the cellular structures.

### In vitro cholesterol trafficking assay

KCs was first treated with NBD-cholesterol for 16 hr, allowing the fluorescent cholesterol analog to integrate into their membranes. Next, KCs were washed with RPMI 1640 and cultured for again 2 hr in the same medium to eliminate any residual fluorescence from the background. These labeled KCs were infected with LD for 4 hr and washed to remove loosely attached MΦs. The KCs were then either immediately fixed with 2% PFA or left untreated to serve as a control group. Finally, the fixed MΦs were examined under a confocal microscope.

### Flow cytometry to determine NBD-cholesterol

KCs were first treated with NBD-cholesterol as previously described. LD parasites were isolated after 2 hr of attachment to the KCs. The quenching of NBD-cholesterol was measured by flow cytometric analysis, using an unstained control.

### TIRF microscopy

To visualize membrane fluidity, TIRF microscopy was performed (*Schirripa Spagnolo and Luin, 2023*). Briefly, KCs were plated on coverslips, and after 24 hr of LD-infection, KCs were fixed with 2% PFA. KCs were stained with Laurdan dye and visualized with Olympus cell TIRF microscopic system. Representative images of three independent biological experiments have been provided.

### RNAseq analysis

RNA-sequencing was performed utilizing external service by Bionivid Project for LD-S and LD-R-infected-PECs, following 24 hr of infection keeping uninfected MΦs as control (Bioproject). Average

expression of differentially expressed genes related to lipid metabolism between LD-S and LD-R-infected-KC is represented as heat map. RAW sequencing reads have been deposited to Indian Nucleotide Data Archive (https://inda.rcb.ac.in/home) with accession number INRP000146.

## Lipid profiling from serum

Lipid profile analysis of patient serum was conducted using lipid estimation kits employing the enzymatic method for the determination of cholesterol, triglycerides, HDL, and LDL. For BALB/c mice, blood was collected following anesthesia and prior to euthanasia. The blood samples were maintained at 4°C to allow clotting, after which the serum was isolated.

## Evaluation of EC50

KCs infected with LD parasites were treated with SAG or Amp-B 24 hr post-infection and incubated for an additional 24 hr. At the experimental endpoints, coverslips were washed with PBS, air-dried, and fixed with 100% ice-cold methanol. Subsequently, the coverslips were stained with Giemsa solution and examined microscopically. To quantify the number of amastigotes, 100 KCs per coverslip were counted (*Mukhopadhyay et al., 2011*). The average of three untreated coverslips was taken as 100% control, and the percentage inhibition of infected KCs in treated cultures was calculated. EC50 values for each isolate were estimated against each drug (*Mukhopadhyay et al., 2011*). Results were also expressed as EC50.

## Aspirin treatment and amastigote load determination

KCs were infected with LD parasites and, after 4 hr, treated with 5 µM Aspirin for a duration of 24 hr. Following this treatment, the KCs were washed and subsequently treated with Amp-B at a specific dose for an additional 24 hr. Coverslips were then washed with PBS, air-dried, and fixed with 100% ice-cold methanol. After fixation, the coverslips were stained with Giemsa solution and examined microscopically. To quantify the number of amastigotes, 100 KCs per coverslip were counted (*Mukhopadhyay et al., 2011*).

## siRNA-mediated KD

For all siRNA transfections, Lipofectamine RNAiMAX Reagent (Life Technologies, 13778100) specifically designed for knockdown assays in primary cells was used according to the manufacturer's instructions with slight modifications. PECs were seeded into 24-well plates at a density of $1 \times 10^5$ per well, and incubated at 37°C with 5% $CO_2$. The transfection complex, comprising (1 µl Lipofectamine RNAiMAX and 50 µl Opti MEM) and (1 µl siRNA and 50 µl Opti MEM) mixed together directly added to the incubated PECs. Gene silencing was checked by IFA and by western blot as mentioned previously.

## Blood samples collection

Blood samples were collected from a total of 22 individuals spanning a diverse age range (8–70 years) by RMRI, Bihar, India. Among these, nine samples were obtained from healthy individuals residing in endemic regions to serve as controls. Serum was isolated from each blood sample through centrifugation, and the lipid profile was subsequently analyzed using a specialized diagnostic kit (Coral Clinical System) following the manufacturer's protocol.

## Statistical analysis

All statistical analyses were performed using GraphPad Prism 8 on raw datasets to ensure robust and reproducible results. For datasets involving comparisons across multiple conditions, one- or two-way ANOVA was conducted, followed by Tukey's post hoc test to assess pairwise differences while controlling for multiple comparisons. A 95% CI was applied to determine the statistical reliability of the observed differences. For non-parametric comparisons across multiple groups, Wilcoxon rank-sum tests were employed, maintaining a 95% CI, which is particularly useful for analysing skewed data distributions. In cases where only two groups were compared, Student's *t*-test was used to determine statistical significance, ensuring an accurate assessment of mean differences. All quantitative data are represented as mean ± SEM to illustrate variability within experimental replicates. Statistical significance was determined at $p \leq 0.05$. Notation for significance levels: *$p \leq 0.05$; **$p \leq 0.001$; ***$p \leq 0.0001$.

## Acknowledgements

SP acknowledges CSIR-UGC (2020–2021) and PMRF (2021–2024) fellowship. DD is a recipient of GATE fellowship. DM is a recipient of CSIR fellowship, SC is a recipient of GATE fellowship. Authors would like to acknowledge IIR lab, SMST, IITKGP for their help in Kupffer cell isolation and flow cytometric analysis, confocal microscope facility under DST-FIST grant conferred on the School of Bioscience, File no. SR/FST/LS-I/2019/595(C), and Confocal facility of Department of Biotechnology, DST-FIST, Govt. of India. Authors would like to thank Prof. Syamal Roy, IICB, India for providing all the LD strains used in this study. Prof. Yasuyuki Goto, The University of Tokyo, Japan for allowing experiments related to live cell video microscopy in his lab. Authors would like to thank Dr. Praphulla Chandra Shukla for providing $Apoe^{-/-}$

mice. Authors acknowledges CRF SMST and CRF IITKGP. Authors would like to thank Dr. Moumita Bhaumik, NICED, India for helping in membrane isolation experiments. Authors would like to thank all the interns of the IDI lab, SMST who help in various experiments related to this project.

## Additional information

### Funding

| Funder | Grant reference number | Author |
|---|---|---|
| University Grants Commission India | CSIR-UGC (2020–2021) fellowship | Supratim Pradhan |
| Prime Minister's Research Fellowship | 2021–2024 | Supratim Pradhan |
| Council of Scientific and Industrial Research | GATE fellowship | Dhruba Dhar Shubhangi Chakraborty |
| Council of Scientific and Industrial Research | CSIR fellowship | Debolina Manna |

The funders had no role in study design, data collection, and interpretation, or the decision to submit the work for publication.

### Author contributions

Supratim Pradhan, Conceptualization, Data curation, Software, Formal analysis, Validation, Investigation, Visualization, Methodology, Writing – original draft, Writing – review and editing; Dhruba Dhar, Debolina Manna, Shubhangi Chakraborty, Arkapriya Bhattacharyya, Khushi Chauhan, Investigation; Rimi Mukherjee, Resources, Investigation; Abhik Sen, Soumen Das, Resources, Writing – review and editing; Krishna Pandey, Resources; Budhaditya Mukherjee, Conceptualization, Resources, Formal analysis, Supervision, Investigation, Visualization, Methodology, Writing – original draft, Project administration, Writing – review and editing

### Author ORCIDs

Supratim Pradhan ⓘ https://orcid.org/0009-0003-6573-8196
Budhaditya Mukherjee ⓘ https://orcid.org/0000-0002-1058-3620

### Ethics

In this study, all procedures involving human participants were conducted in accordance with the ethical standards of the 1964 Helsinki Declaration and approval of the 'Institutional Human Ethical Committee' of ICMR-Rajendra Memorial Research Institute of Medical Sciences, Patna, India (Approval No. RMRI/EC/20/2020). Following information about the potential risks, benefits, and the investigational nature of the study consent from all study participants was obtained in the informed consent form.

All experiments were conducted in accordance to Institutional Animal Ethics Committee (IAEC), IIT Kharagpur guidelines India (IAEC/BI/1 55/2021) and approved by National Regulatory Guidelines issued by Committee for the Purpose of Supervision of Experiments on Animals, Ministry of Environment and Forest, Government of India.

Reviewer #1 (Public review): https://doi.org/10.7554/eLife.102857.3.sa1
Author response https://doi.org/10.7554/eLife.102857.3.sa2

## Additional files

### Supplementary files
MDAR checklist

### Data availability
RAW sequencing reads have been deposited to Indian Nucleotide Data Archive (https://inda.rcb.ac.in/home) with accession number INRP000146 (https://ibdc.rcb.res.in/inda/completeStudyDetailsById?studyid=INRP000146).

The following dataset was generated:

| Author(s) | Year | Dataset title | Dataset URL | Database and Identifier |
|---|---|---|---|---|
| Pradhan S, Dhar D, Manna D, Bhattacharyya A, Das S, Mukherjee B | 2025 | Differential gene expression analysis of Leishmania donovani infected macophages In a time dependent manner | https://ibdc.rcb.res.in/inda/completeStudyDetailsById?studyid=INRP000146 | inda, INRP000146 |

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
