## [Editor Report · eLife Assessment]

This **important** study investigates the propensity of the intravacuolar pathogen, Leishmania, to scavenge lipids which it utilizes for its accelerated growth within macrophages. The authors present **compelling** evidence that supports this hypothesis, although the genetic basis for the parasite's requirement for lipids remains unresolved. The study adds to other work that has implicated pathogen-derived processes in the selective recruitment of vesicles to the pathogen-containing vacuole, based on the content of the cargo.

---

## [Referee Report · Reviewer #1 (Public review)]

Although the use of antimony has been discontinued in India, the observation that Leishmania parasites resistant to antimony are in circulation has been cited as evidence that these resistant parasites are now a distinct strain with properties that ensure their transmission and persistence. It is of interest to determine the properties that favor the retention of their drug resistance phenotype even in the absence of the selective pressure that the drug would otherwise exert. The hypothesis that these authors set out to test is that these parasites have developed a new capacity to acquire and utilize lipids, especially cholesterol, which enables them to grow robustly in infected hosts. The authors present compelling evidence that supports their hypothesis. However, the genetic basis for the parasite's gluttony for lipids remains unresolved.

An issue raised in the initial review was the insufficient detail in the discussion of experiments in which parasitophorous vacuoles were isolated from infected cells and their molecular content was investigated. In this new version, the authors have provided more details of those experiments, including the relative enrichment of the preparations.

A puzzling observation for which they provide compelling evidence is the capacity of LD-R parasites to undergo logarithmic growth between 4 and 24 hours in infected cells. Interestingly, after logarithmic growth within the macrophages at those early times, parasite growth slows down inexplicably after 24 hours. One is left to imagine what the consequences of such growth within an infected host will be.

They made the novel observation that Lamp1 expression increases as early as 4 hours after infection with LD-R and by 12 hours after infection with both LD-S and LD-R. Interestingly, transcription analysis did not provide evidence for increased Lamp1 transcripts, leaving open the possibility that parasite infection may exert control over host cell biology at the translational level. How this might be achieved and what genes would be targets for such controls are open questions.

The dynamic changes in lipid biosynthesis pathways, in comparison to lipid uptake and the formation of lipid droplets in infected cells, are tracked. The basis for the beneficial effect of aspirin to stifle parasite resistance to antimony is explored.

---

## [Author Response]

The following is the authors’ response to the original reviews

**Public Reviews:**

**Reviewer #1 (Public review):**
Although the use of antimony has been discontinued in India, the observation that there are Leishmania parasites that are resistant to antimony in circulation has been cited as evidence that these resistant parasites are now a distinct strain with properties that ensure their transmission and persistence. It is of interest to determine what are the properties that favor the retention of their drug resistance phenotype even in the absence of the selective pressure that would otherwise be conferred by the drug. The hypothesis that these authors set out to test is that these parasites have developed a new capacity to acquire and utilize lipids, especially cholesterol which affords them the capacity to grow robustly in infected hosts.

We sincerely appreciate Reviewer 1's thoughtful and positive evaluation of our manuscript. We acknowledge that the reviewer has a few major concerns, and we would like to address them one by one in the following section.

Major issues:(1) There are several experiments for which they do not provide sufficient details, but proceed to make significant conclusions.Experiments in section 5 are poorly described. They supposedly isolated PVs from infected cells. No details of their protocol for the isolation of PVs are provided. They reference a protocol for PV isolation that focused on the isolation of PVs after L. amazonensis infection. In the images of infection that they show, by 24 hrs, infected cells harbor a considerable number of parasites. Is it at the 24 hr time point that they recover PVs? What is the purity of PVs? The authors should provide evidence of the success of this protocol in their hands. Earlier, they mentioned that using imaging techniques, the PVs seem to have fused or interconnected somehow. Does this affect the capacity to recover PVs? If more membranes are recovered in the PV fraction, it may explain the higher cholesterol content.

We would like to thank the reviewer for correctly pointing out lack of details regarding PV isolation and its purity. There are multiple questions raised by the reviewer and we will answer them one by one in a point wise manner:

Firstly, “Is it at the 24 hr time point that they recover PVs?”

In the ‘Methods’ section of the original submission (Line number 606-611), there is a separate section on “Parasitophorous vacuole (PV) Isolation and cholesterol measurement”, where it is clearly mentioned, “24Hrs LD infected KCs were lysed by passing through a 22-gauge syringe needle to release cellular contents. Parasitophorous vacuoles (PV) were then isolated using a previously outlined protocol [Ref: 73].” However, we do acknowledge further details might be useful to enrich this section, and hence we would like to include the following details in the Methods section of the revised manuscript, Line 663-678 “Parasitophorous vacuoles (PV) were isolated using a previously outlined protocol with slight modifications [76]. 107 KCs were seeded in a 100 mm plate and allowed to adhere for 24Hrs. Following this infection was performed with *Leishmania donovani* (LD) for 24Hrs, the infected KCs were then harvested by gentle scraping and lysed through five successive passages through an insulin needle to ensure membrane disruption while preserving organelle integrity. The lysate was centrifuged at 200 × g for 10mins at 4°C to remove intact cells and large debris. The resulting supernatant was carefully collected and subjected to a discontinuous sucrose density gradient (60%, 40%, and 20%). The gradient was centrifuged at 700 × g for 25mins at 4°C to facilitate organelle separation. The interphase between the 40% and 60% sucrose layers, enriched with PVs, was carefully collected and subjected to a final centrifugation step at 12,000 × g for 25mins at 4°C. The supernatant was discarded, and the resulting pellet was enriched for purified parasitophorous vacuoles, suitable for downstream biochemical and molecular analyses. Cholesterol and protein contents in PV were determined by an Amplex Red assay kit and Bradford assay, respectively. Resulting data were represented as micrograms of cholesterol per microgram of protein.”

Secondly, What is the purity of PVs? Earlier, they mentioned that using imaging techniques, the PVs seem to have fused or interconnected somehow. Does this affect the capacity to recover PVs? If more membranes are recovered in the PV fraction, it may explain the higher cholesterol content.

We appreciate the reviewer for pointing this critical lack of data in the submitted manuscript. In the revised manuscript, we have now provided data on the purity of isolated fraction by performing Confocal imaging and Western blot against PV and cytoplasmic fraction in the revised manuscript. We admit, as rightly pointed out by the reviewer we need to access the purity of isolated PV in our experiment. As suggested by the reviewer, we have included the results of this experiment in the Figure 3C i, C ii and C iii. Our results clearly showed an efficient PV isolation with demarcating LAMP-1 positive staining around LD amastigotes, which was further validated by Western Blot showing a significant enrichment of LAMP-1 specifically in the PV fraction. This has been included as (Line 225-234), in the revised manuscript which read as, “Parasitophorous vacuole fractions were isolated from LD-S and LD-R-infected KCs at 24Hrs p.i. using a previously established protocol [35]. Following isolation, PV purity was confirmed through LAMP-1 staining which showed a significant enrichment around isolated PV in Confocal microscopy (Figure 3C i). Purity of isolated PV fractions was further confirmed by Western blot which showed an enhanced enrichment of LAMP-1 for LD-R-PV fraction as compared to LD-S-PV fraction, while PV excluded cellular fraction showed residual LAMP-1 expression confirming the purity of the isolated PV fractions (Figure 3C ii, iii). Following isolation, protein concentration was measured for isolated PV fractions using the Bradford assay, and PV fractions from both LD-S- and LD-R-infected KCs were normalized accordingly.”

(2) In section 6 they evaluate the mechanism of LDL uptake in macrophages. Several approaches and endocytic pathway inhibitors are employed. The authors must be aware that the role of cytochalasin D in the disruption of fluid phase endocytosis is controversial. Although they reference a study that suggests that cytochalasin D has no effect on fluid-phase endocytosis, other studies have found the opposite (doi: 10.1371/journal.pone.0058054). It wasn't readily evident what concentrations were used in their study. They should consider testing more than 1 concentration of the drug before they make their conclusions on their findings on fluid phase endocytosis.

We thank the reviewer for this insightful comment and we apologise for missing out mentioning Cytochalasin-D concentration. To clarify, LDL uptake by LD-R infected KCs is LDL-receptor independent as clearly shown in Section 6, Figure 4A, Figure S4A, Figure S4B i and Figure S4B ii in the submitted manuscript. In (Figure 4F and Figure S4D) of the submitted manuscript, as referred by the Reviewer, Cytochalasin-D was used at a concentration of 2.5µg/ml. At this concentration, we did not observe any effect of Cytochalasin-D on LDL-receptor independent fluid phase endocytosis as intracellular LD-R amastigotes was able to uptake LDL successfully and proliferate in infected Kupffer cells, unlike Latranculin-A (5µM) treatment which completely inhibited intracellular proliferation of LD-R amastigotes by blocking only receptor independent Fluid phase endocytosis (Video 2A and 2B and Figure 4E in the submitted manuscript). In fact, the study referred by the reviewer (doi: 10.1371/journal.pone.0058054), used a concentration of 4µg/ml Cytochalasin-D which did affect both LDL-receptor dependent and also receptor independent endocytosis in bone marrow derived macrophages. We would also like to clarify that in this work during our preliminary experiments we have also tested higher concentration Cytochalasin-D (5µg/ml). However, even at this higher concentration there were no significant effect of Cytochalasin-D on LD-R induced LDL-receptor independent fluid phase endocytosis as observed from intracellular LD-R amastigote count. Thus, we strongly believe that Cytochalasin-D does not have any impact on LD-R induced fluid phase endocytosis even at higher concentration. We have now included this data as Figure 4F and Figure S4E in the revised manuscript. Further, to clear out any confusion that readers might have, and also concentration of all the inhibitors used in the study will be mentioned in the Result section (Line 278 and 284), as well as in the revised Figure labels.

(3) In Figure 5 they present a blot that shows increased Lamp1 expression from as early as 4 hrs after infection with LD-R and by 12 hrs after infection of both LD-S and LD-R. Increased Lamp1 expression after Leishmania infection has not been reported by others. By what mechanism do they suggest is causing such a rapid increase (at 4hrs post-infection) in Lamp-1 protein? As they report, their RNA seq data did not show an increase in LAMP1 transcription (lines 432-434).

We would like to express our gratitude to the reviewer for highlighting the novelty of this observation. Indeed, to the best of our knowledge, no similar findings (we could not find reference of any quantitative Western blot for LAMP-1) have been reported previously in primary macrophages infected with *Leishmania donovani* (LD). Firstly, we would like to point out, as stated in the Methods section (Lines 556–566) of the submitted manuscript: "Flow-sorted metacyclic LD promastigotes were used at a MOI of 1:10 (with variations of 1:5 and 1:20 in some cases) for 4 hours, which was considered the 0th point of infection. Macrophages were subsequently washed to remove any extracellular loosely attached parasites and incubated further as per experimental requirements.” This indicates that our actual study points correspond to approximately the 8th and 28th hours post-infection”. We just wanted to clarify the time point just to prevent any potential confusion.

Now regarding LAMP1 expression, although we could not find any previous reports of its expression in LD infected primary macrophages, we would like to mention that there is a previous report (doi.org/10.1128/mBio.01464-20), which shows a similar punctuated LAMP-1 upregulation (as observed by us in Figure 5A i of the submitted manuscript) in response to leishmania infection in nonphagocytic fibroblast. It is tempting to speculate that increased LAMP-1 expression observed in response to LD-R infected macrophages might be due to increased lysosomal biogenesis, required for degrading increased endocytosed-LDL into bioavailable cholesterol. However, since no change in LAMP-1 expression in RNA seq data (Figure 6, of the submitted manuscript), we can only speculate that this is happening due to some post transcriptional or post translational modifications. But further work will definitely require to investigate this mechanism in details which is beyond the scope of this work. That is why, in the submitted manuscript, (Line 432-435), we have discussed this, “Although available RNAseq analysis (Figure 6) did not support this increased expression of lamp-1 in the transcript level, it did reflect a notable upregulation of vesicular fusion protein (VSP) vamp8 and stx1a in response to LD-R-infection. LD infection can regulate LAMP-1 expression, and the role of VSPs in LDLvesicle fusion with LD-R-PV is worthy of further investigation.”

However, we agree with the reviewer that this might not be enough for the clarification. Hence in the revised manuscript this has been updated in the Discussion section (Line 465-472) as follows, “Although available *RNAseq* analysis (Figure 6) did not support this increased expression of *lamp-1* in the transcript level, it did reflect a notable upregulation of vesicular fusion protein (VSP) *vamp8* and *stx1a* in response to LD-R-infection. How, LD infection can regulate LAMP-1 expression, and the role of VSPs in LDL-vesicle fusion with LD-R-PV is worthy of further investigation. It is possible and has been earlier reported that LD infection can regulate host proteins expression through post transcriptional and post translational modifications [61-63]. It is tempting to speculate that LD-R amastigote might be promoting an increased lysosomal biogenesis through any such mechanism to increase supply of bioavailable cholesterol through action of lysosomal acid hydrolases on LDL.”

(4) In Figure 6, amongst several assays, they reported on studies where SPC-1 is knocked down in PECs. They failed to provide any evidence of the success of the knockdown, but nonetheless showed greater LD-R after NPC-1 was knocked down. They should provide more details of such experiments.

Although we do understand the concern raised by the reviewer, this statement in question is factually incorrect. We would like to point out that in Figure 6F i, of the submitted manuscript (Figure 6G ii in the revised manuscript), we have demonstrated decreased NPC-1 staining following transfection with NPC-1-specific siRNA, whereas no such reduction was observed with scrambled RNA. Similar immunofluorescence data confirming LDL-receptor knockdown has also been provided in Figure S4B i of the submitted manuscript (Figure S4B ii in the revised manuscript). However, we acknowledge that the reviewer may be referring to the lack of quantitative validation of the knockdown via Western blot. We would like to clarify although, we already had this data, but we did not include it to avoid duplication to reduce the data density of the MS. But as suggested by the reviewer, we have included western blot for both NPC-1 and LDL-receptor knock down in the revised manuscript as Figure 6G i and Figure S4B i which again confirms an efficient Knock down of NPC-1 and LDLr as we have observed with IFA.

Additionally, as suggested by the reviewer, we also noticed lack of details in Methods section of the submitted manuscript, concerning siRNA mediated Knock down (KD). Therefore, we have included more details in the revised manuscript (Line 821-828), which read as, “For all siRNA transfections, Lipofectamine RNAiMAX Reagent (Life Technologies, 13778100) specifically designed for knockdown assays in primary cells was used according to the manufacturer's instructions with slight modifications. PECs were seeded into 24-well plates at a density of 1x10^5^ per well, and incubated at 37°C with 5% CO2. The transfection complex, comprising (1µl Lipofectamine RNAiMAX and 50µl Opti MEM) and (1 µl siRNA and 50µl Opti MEM) mixed together directly added to the incubated PECs. Gene silencing was checked by IFA and by Western blot as mentioned previously.”

Minor issues(1) There is an implication that parasite replication occurs well before 24hrs post-infection?Studies on Leishmania parasite replication have reported on the commencement of replication after 24hrs post-infection of macrophages (PMCID: PMC9642900). Is this dramatic increase in parasite numbers that they observed due to early parasite replication?

We thank the reviewer for this insightful comment and appreciate the opportunity to clarify our findings. Indeed, as rightly assumed by the Reviewer, as our data suggest, and we also believe that this increase intracellular amastigotes number is a consequence of early replication of *Leishmania donovani*. As already mentioned in response to Point number 3 raised by Reviewer 1, we would again like to highlight that in the Methods section (Lines 562–566), it is clearly stated: "Flow-sorted metacyclic LD promastigotes were used at a MOI of 1:10 (with variations of 1:5 and 1:20 in some cases) for 4 hours, which was considered the 0th point of infection. Macrophages were subsequently washed to remove any extracellular loosely attached parasites and incubated further as per experimental requirements.” This effectively means that our actual study points correspond to approximately the 8th and 28th hours post-infection and we just want to mention it to avoid any confusion regarding experimental time points.

Now, regarding specific concern related to Leishmania parasite replication, we would like to point out that the study referred by the reviewer on the commencement of replication after 24hrs, was conducted on *Leishmania major*, which may differ significantly from *Leishmania donovani* owing to its species and strain-specific characteristics (PMCID: PMC9642900). In fact, doubling time of *Leishmania donovani* (LD) has been previously reported to be approximately 11.4 hours (doi: 10.1111/j.1550-7408. 1990.tb01147.x). Moreover, multiple studies have indicated an exponential increase in intracellular LD amastigote number (more than two-fold increase) by 24Hrs post infection. (doi:10.1128/AAC.0119607, doi.org/10.1016/j.ijpara.2011.07.013). We also have a similar observation for both infected PEC and KC as depicted in Figure 1C and Figure S1C in the submitted and revised manuscript indicating that active replication is happening in this time frame for *Leishmania donovani*. Hence it was an informed decision from our side to focus on 24Hrs time point to perform the analysis on intracellular LD proliferation.

(2) Several of the fluorescence images in the paper are difficult to see. It would be helpful if a blown-up (higher magnification image of images in Figure 1 (especially D) for example) is presented.

We apologise for the inconvenience. Although we have provided Zoomed images for several other Figures in the submitted manuscript and revised manuscript, like Figure 4, Figure 5, Figure 6 and Figure 8. However, this was not always doable for all the figures (like for Figure 1D), due to lack of space and Figure arrangements requirements. However, to accommodate Reviewer’s request we have provide a blown-up image for Figure 1D iii in the revised manuscript.

(3) The times at which they choose to evaluate their infections seem arbitrary. It is not clear why they stopped analysis of their KC infections at 24 hrs. As mentioned above, several studies have shown that this is when intracellular amastigotes start replicating. They should consider extending their analyses to 48 or 72 hrs post-infection. Also, they stop in vitro infection of Apoe/- mice at 11 days. Why? No explanation is given for why only 1 point after infection.

Reviewer has raised two independent concerns and we would like to address them individually.

Firstly, “The times at which they choose to evaluate their infections seem arbitrary. It is not clear why they stopped analysis of their KC infections at 24 hrs. As mentioned above, several studies have shown that this is when intracellular amastigotes start replicating. They should consider extending their analyses to 48 or 72 hrs post-infection.”

We have already provided a detail justification for time point selection in our response to Reviewer 1, Minor Comment 1. As mentioned already we observed a significant and sharp rise in the number of intracellular amastigotes between 4Hrs and 24Hrs post-infection in KC, with replication rate appeared to be not increasing proportionally (not doubling) after that (Figure 1C in the revised manuscript). This early stage of rapid replication of LD amastigotes, therefore likely coincides with a critical period of lipid acquisition by intracellular amastigotes (Video 3A and 3B and Figure 4E in the submitted manuscript and revised manuscript) and thus 24Hrs infected KC was specifically selected. In this regard, we would further like to add that at 72Hrs post-infection, we noticed a notable number of infected Kupffer cells began detaching from the wells with extracellular amastigotes probably egressing out. This phenomenon potentially reflects the severe impact of prolonged infection on Kupffer cell viability and adhesion properties as shown in Video 2 in the revised manuscript and Author response image 1. This observation further influenced our decision to conclude all infection studies in Kupffer cells by the 48Hrs post-infection, which necessitate to complete the infection time point at 24 Hrs, for allowing treatment of Amp-B for another 24 Hrs (Figure 8, and Figure S5, in the submitted manuscript and revised manuscript). We acknowledge that we should have been possibly clearer on our selection of infection time points and as the Reviewer have suggested we have included this information in the revised manuscript (Line 134-141) for clear understanding of the reader. This read as, “Interestingly, as compared to a significant and sharp rise in the number of intracellular amastigotes between 4Hrs and 24Hrs post infected KC in response to LD-R infection, the number of intracellular amastigotes although increased significantly did not doubled from 24Hrs to 48Hrs p.i. suggesting exponential LD amastigote replication between 4Hrs and 24Hrs time frame and slowing down after that (Figure 1Ci, ii). Moreover, it was also noticed that at 72Hrs p.i. a notable number of infected-KC began detaching from the wells with extracellular amastigotes probably egressing out from the infected-KCs (Video 2). Thus, 24Hrs time point was selected to conduct all further infection studies involving KCs.”

**Author response image 1. sa2fig1:** Representative images of Kupffer cells infected with *Leishmania donovani* at 72Hrs post-infection showing a significant morphological change. Infected cells exhibit a rounded morphology and progressive detachment. Scale bar 10µm.

Secondly “Also, they stop in vitro infection of Apoe-/- mice at 11 days. Why? No explanation is given for why only 1 point after infection.”

We apologize for not providing an explanation regarding the selection of the 11-day time point for *Apoe-/-* experiments (Figure 2 of the submitted and revised manuscript). Our rationale for this choice is based on both previous literature and the specific objectives of our study. Previous report suggests that *Leishmania donovani* infection in hypercholesteraemic *Apoe-/-* mice triggers a heightened inflammatory response at approximately six weeks’ post-infection compared to C57BL/6 mice, leading to more efficient parasite clearance. This is owing to unique membrane composition of Apoe^-/-^ which rectifies leishmania mediated defective antigen presentation at a later stage of infection (DOI 10.1194/jlr.M026914). Additionally, previous studies have also indicated that *Leishmania donovani* infection is well-established in vivo within 6 to 11 days post-infection in murine models (doi: 10.1128/AAC.47.5.1529-1535.2003). Given that in this experiment we particularly aimed to assess the early infection status (parasite load) in diet-induced hypercholesterolemic mice, we would like to argue that the selection of the 11-day time point was rational and well-aligned with our study objectives as this time point within this window are optimal for capturing initial parasite burden depending on initial lipid utilization, before host-driven immune clearance mechanisms could significantly alter infection dynamics. We have included this explanation in the revised manuscript (Line 170-179) as suggested by the Reviewer and this read as, “Previous report has suggested that LD infection in hypercholesteremic Apoe^-/-^ mice triggers a heightened inflammatory response at approximately six weeks’ post-infection compared to wild type BL/6 mice, leading to more efficient parasite clearance. This is owing to unique membrane composition of Apoe-/- which rectifies leishmania mediated defective antigen presentation at a later stage of LD infection [20]. Additionally, previous studies have also indicated that LD infection is well-established in mice within 6 to 11 days post-infection in murine models [33]. Thus to evaluate impact of initial lipid utilization on LD amastigote replication in vivo, BL/6 and diet-induced hypercholesterolemic Apoe^-/-^ mice were infected with GFP expressing LD-S or LD-R promastigotes and sacrificed 11 days p.i.”

**Reviewer #2 (Public review):**
Summary:This study by Pradhan et al. offers critical insights into the mechanisms by which antimonyresistant Leishmania donovani (LD-R) parasites alter host cell lipid metabolism to facilitate their own growth and, in the process, acquire resistance to amphotericin B therapy. The authors illustrate that LD-R parasites enhance LDL uptake via fluid-phase endocytosis, resulting in the accumulation of neutral lipids in the form of lipid droplets that surround the intracellular amastigotes within the parasitophorous vacuoles (PV) that support their development and contribute to amphotericin B treatment resistance. The evidence provided by the authors supporting the main conclusions is compelling, presenting rigorous controls and multiple complementary approaches. The work represents an important advance in understanding how intracellular parasites can modify host metabolism to support their survival and escape drug treatment.

We would like to sincerely thank the reviewer for appreciating our work and find the evidence compelling to address the issue of emergence of drug resistance in infection with intracellular protozoan pathogens.

Strengths:(1) The study utilizes clinical isolates of antimony-resistant L. donovani and provides interesting mechanistic information regarding the increased LD-R isolate virulence and emerging amphotericin B resistance.(2) The authors have used a comprehensive experimental approach to provide a link between antimony-resistant isolates, lipid metabolism, parasite virulence, and amphotericin B resistance. They have combined the following approaches:a) In vivo infection models involving BL/6 and Apoe-/- mice.b) Ex-vivo infection models using primary Kupffer cells (KC) and peritoneal exudate macrophages (PEC) as physiologically relevant host cells.c) Various complementary techniques to ascertain lipid metabolism including GC-MS, Raman spectroscopy, microscopy.d) Applications of genetic and pharmacological tools to show the uptake and utilization of host lipids by the infected macrophage resident L. donovani amastigotes.(3) The outcome of this study has clear clinical significance. Additionally, the authors have supported their work by including patient data showing a clear clinical significance and correlation between serum lipid profiles and treatment outcomes.(4) The present study effectively connects the basic cellular biology of host-pathogen interactions with clinical observations of drug resistance.(5) Major findings in the study are well-supported by the data:a) Intracellular LD-R parasites induce fluid-phase endocytosis of LDL independent of LDL receptor (LDLr).b) Enhanced fusion of LDL-containing vesicles with parasitophorous vacuoles (PV) containing LD-R parasites both within infected KCs and PECs cells.c) Intracellular cholesterol transporter NPC1-mediated cholesterol efflux from parasitophorous vacuoles is suppressed by the LD-R parasites within infected cells.d) Selective exclusion of inflammatory ox-LDL through MSR1 downregulation.e) Accumulation of neutral lipid droplets contributing to amphotericin B resistance.Weaknesses:The weaknesses are minor:(1) The authors do not show how they ascertain that they have a purified fraction of the PV postdensity gradient centrifugation.(2) The study could have benefited from a more detailed analysis of how lipid droplets physically interfere with amphotericin B access to parasites.

We have addressed both these concerns in the revised Version of this work as elaborated in the following section.

Impact and significance:This work makes several fundamental advances:(1) The authors were able to show the link between antimony resistance and enhanced parasite proliferation.(2) They were also able to reveal how parasites can modify host cell metabolism to support their growth while avoiding inflammation.(3) They were able to show a certain mechanistic basis for emerging amphotericin B resistance.(4) They suggest therapeutic strategies combining lipid droplet inhibitors with current drugs.
**Recommendations for the authors:**

**Reviewer #2 (Recommendations for the authors):**
(1) Experimental suggestions:a) The authors could have provided a more detailed analysis of lipid droplet composition. This is a critically missing piece in this nice study.

We completely agree with the Reviewer on this, a more detailed analysis of lipid droplets composition, dynamics of its formation and mechanism of lipid transfer to amastigotes residing within the PV would be worthy of further investigation. To answer the Reviewer, we are already conducting investigation in this direction and have very promising initial results which we are willing to share with the Reviewer as unpublished communication if requested. Since, we plan to address these questions independently, we hope Reviewer will understand our hesitation to include these data into the present work which is already data dense. We sincerely believe existence of lipid droplet contact sites with the PV along with the specific lipid type transfer to amastigotes and its mechanism requires special attention and could stand out as an independent work by itself.

b) The macrophages (PEC, KC) could have been treated with latex beads as a control, which would indicate that cholesterol and lipids are indeed utilized by the Leishmania parasitophorous vacuole (PV) and essential for its survival and proliferation.

We thank the reviewer for this nice suggestion, which we believe will further strengthen the conclusion of this work. We have now included this data as Figure 5E in the revised manuscript. Our data showed that infected KC harbouring both LD-R amastigotes and Fluorescent Latex Beads, showed a concentrated staining of Cholesterol around amastigotes, with no positive Cholesterol staining around internalized latex beads similar to LD-S amastigotes. This observation clearly confirmed specific lipid uptake in LD-R-PV, which can not be replicated by phagocytosed Latex Beads.

c) HMGCoA reductase is an important enzyme for the mevalonate pathway and cholesterol synthesis. The authors have not commented on this enzyme in either host or parasite. Additionally, western blots of these enzymes along with SREBP2 could have been performed.

We appreciate the concern and do see the point why reviewer is suggesting this. We would like to mention that regarding HMGCoA we already do have real time qPCR data which perfectly aligns with our RNAseq data (Figure 6 A i, in the submitted and revised manuscript), showing significant downregulation specifically in LD-R infected KC as compared to uninfected control. We are including this data as Author response image 2. However, we did not proceed with checking the level of HMGCoA at the protein level as we noticed several previous reports have suggested that HMGCoA reductase remains under transcriptional control of SERBP2 (doi.org/10.1016/j.cmet.2011.03.005, doi: 10.1194/jlr.C066712, doi:10.1194/jlr.RA119000201), which acts the master regulator of mevalonate pathway and cholesterol synthesis (doi.org/10.1161/ATVBAHA.122.317320) and SERBP2 remains significantly downregulated in response to LD-R infection (Figure 6B i and Figure 6C in the submitted and revised manuscript). However, as suggested by the Reviewer, we have updated this data in the revised manuscript as Figure 6D. Western blot data further confirmed a significant expected downregulation of HMGCoA in response to LD-R infection.

**Author response image 2. sa2fig2:** qPCR Analysis of HMGCR Expression Following *Leishmania donovani* Infection: Quantitative PCR analysis showing the relative expression of *hmgcr* (3-hydroxy-3-methylglutaryl-CoA reductase) in Kupffer cells after 24 hours of *Leishmania donovani* (LD) infection compared to uninfected control cells. Gene expression levels are normalized to *β-actin* as an internal control, and fold change is represented relative to the uninfected condition.

d) The authors should discuss the expression pattern of any enzyme of the mevalonate pathway that they have found to be dysregulated in the transcript data.

As per the reviewer’s suggestion, we have looked into the *RNA seq* data and observed that apart from *hmgcr*, *hmgcs* (3-hydroxy-3methylglutaryl-CoA synthase), another key enzyme in the mevalonate pathway, is significantly downregulated in host PECs in response to LD-R infection compared to the LD-S infection. We have Discussed this in the revised manuscript (Line 484-490), which read as “Further RNA sequencing data also revealed a significant downregulation of *hmgcs* (3-hydroxy-3-methylglutarylCoA synthase) in LD-R infected PECs as compared to LD-S infection. Downregulation of HMGCS which catalyzes the condensation of acetyl-CoA with acetoacetyl-CoA to form 3-hydroxy-3-methylglutaryl-CoA (HMG-CoA), which serves as an intermediate in both cholesterol biosynthesis and ketogenesis further supports our observation that LD-R-infected PECs preferentially rely on endocytosed low-density lipoprotein (LDL)-derived cholesterol rather than de novo synthesized cholesterol to support their metabolic needs.”

e) The authors have followed a previously published protocol by Real F (reference 73) to enrich for parasitophorous vacuole (PV). However, they do not show how they ascertain that they have a purified fraction of the PV post-density gradient centrifugation. The authors should at least show Western blot data for LAMP1 for different fractions of density gradient from which they enriched the PV.

As we previously stated in our response to Reviewer 1, in the revised manuscript we have included a detailed analysis of purity for different fractions during PV isolation. We sincerely appreciate the reviewer for highlighting this important concern and for suggesting an approach to conduct the experiment. We have included this data as Figure 3C i, ii, iii in the revised manuscript. Our Imaging and Western blot data showed a significant enrichment of LAMP-1 in PV fraction, and we believe this result further reinforce the conclusions of our study on increased Cholesterol.

(2) Presentation improvements:a) Add a clear timeline for infection experiments.

As suggested by the Reviewer, we have included a schematic of Timelines for all the animal infection experiment (Figure 2Ci and Figure 7A,Fi) in the revised manuscript.

b) Provide more details on patient sample collection and analysis.

We have included more details on the sample collection in the Method section of the revised manuscript (Line 830-835), “Blood samples were collected from a total of 22 individuals spanning a diverse age range (8 to 70 years) by RMRI, Bihar, India. Among these, nine samples were obtained from healthy individuals residing in endemic regions to serve as controls. Serum was isolated from each blood sample through centrifugation, and the lipid profile was subsequently analysed using a specialized diagnostic kit (Coral Clinical System) following the manufacturer's protocol.”

c) Consider reorganizing figures to better separate mechanistic and clinical findings.

We would like to thank the reviewer for this suggestion. We felt that a major arrangement altering the sequence of the Figures as presented in the Original Submission will impact smooth flow of the story and hence, we did not disturb that. However, as suggested by the Reviewer we have performed major rearrangement within Figure 2, Figure 5 and Figure 6 and Figure 9 of the revised manuscript for a better representation of the data and convenience of the reader. Also, if the reviewer has specific suggestion regarding rearrangement of any particular figure, we will be happy to consider that.

(3) Technical clarifications needed:a) Specify exact concentrations used for inhibitors.

We apologise for this unwanted and unnecessary mistake. Please note we have now clearly mentioned the concentration of all the inhibitors used in this study in Result section and in the Figures of the revised manuscript. For easy understanding The revised section (Line 281-287) read as, “Finally, we infected the KCs with GFP expressing LD-R for 4Hrs, washed and allowed the infection to proceed in presence of fluorescent red-LDL and Latrunculin-A (5µM), a compound which specifically inhibits fluid phase endocytosis by inducing actin depolymerization [41]. Real-time fluorescence tracking demonstrated that Latrunculin-A treatment not only prevented the uptake of fluorescent red-LDL but also severely impacted intracellular proliferation of LD-R amastigotes (Video 2A and 2B and Figure 4E). In contrast, treatment with Cytochalasin-D, which alters cellular F-actin organization but does not affect fluid phase endocytosis [41], had no effect on the intracellular proliferation of LD-R irrespective of Cytochalasin-D concentrations (2.5µg/ml and 5µg/ml respectively) (Figure 4F and Figure S4D).”

b) Include more details on image analysis methods.

Please note that in specific sections like in Line numbers 574-579, 653-658, 10471049 of the submitted manuscript, we have put special attention in describing the Image analysis process. However, we agree that in some particular cases more details will be appreciated by the reader. Hence, we have included an additional section of Image Analysis in the Methods section of the revised manuscript. This section (Line 727-739) read as, “Image processing and analysis were conducted using Fiji (ImageJ). For optimal visualization, Giemsa-stained macrophages (MΦs) were represented in grayscale to enhance contrast and structural clarity. To improve the distinction of different fluorescent signals, pseudo-colors were assigned to fluorescence images, ensuring better differentiation between various cellular components. For colocalization analysis (Figures 3, Figure 5, Figure 6, and Figure S2), we utilized the RGB profile plot plugin in ImageJ, which allows for the precise assessment of signal overlap by generating fluorescence intensity profiles across selected regions of interest. This approach provided quantitative insights into the spatial relationship between labelled molecules within infected cells. Additionally, for analyzing the distribution of cofilin in Figure 4, the ImageJ surface plot plugin was employed. This tool enabled three-dimensional visualization of fluorescence intensity variations, facilitating a more detailed examination of cofilin localization and its potential reorganization in response to infection.”

c) Clarify statistical analysis procedures.

We have already provided a dedicated section of Statistical Analysis in the Methods section of the Original Submission and also have also shown the groups being compared to determine the statistical analysis in the Figure and in the Figure Legends of the submitted manuscript. Furthermore, as suggested by the Reviewer we have now also add additional clarification regarding the statistical analysis performed in the revised manuscript (Line 737-749). In the revised manuscript this section read as, “All statistical analyses were performed using GraphPad Prism 8 on raw datasets to ensure robust and reproducible results. For datasets involving comparisons across multiple conditions, one-way or two-way analysis of variance (ANOVA) was conducted, followed by Tukey’s post hoc test to assess pairwise differences while controlling for multiple comparisons. A 95% confidence interval (CI) was applied to determine the statistical reliability of the observed differences. For non-parametric comparisons across multiple groups, Wilcoxon rank-sum tests were employed, maintaining a 95% confidence interval, which is particularly useful for analysing skewed data distributions. In cases where only two groups were compared, Student’s t-test was used to determine statistical significance, ensuring an accurate assessment of mean differences. All quantitative data are represented as mean ± standard error of the mean (SEM) to illustrate variability within experimental replicates. Statistical significance was determined at P ≤ 0.05. Notation for significance levels: *P ≤ 0.05; **P ≤ 0.001; ***P ≤ 0.0001.”

(4) Minor corrections:a) Methods section could benefit from more details on Raman spectroscopy analysis.

We agree with this suggestion of the Reviewer. For providing more clarity have incorporate additional details in the Methodology for the Raman section of the revised manuscript (Line 638-649). The updated section will read as follows in the revised manuscript. “For confocal Raman spectroscopy, spectral data were acquired from individual cells at 1000× magnification using a 100 × 100 μm scanning area, following previously established specifications. After spectral acquisition, distinct Raman shifts corresponding to specific biomolecular signatures were extracted for further analysis. These included: Cholesterol (535–545 cm¹), Nuclear components (780–790 cm¹), Lipid structures (1262–1272 cm^1^), Fatty acids (1436–1446 cm^1^) Following spectral extraction, pseudo-color mapping was applied to highlight the spatial distribution of each biomolecular component within the cell. These processed spectral images are presented in Figure 3D1, where the first four panels illustrate the individual biomolecular distributions. A merged composite image was then generated to visualize the co-localization of these biomolecules within the cellular microenvironment, with the final panel specifically representing the spatial distribution of key biomolecules.”

b) In the methods section line 609, page 14, the authors cite Real F protocol as reference 73 for PV enrichment. However, in the very next section on GC-MS analysis (lines 615-616, page 15), they state they have used reference 74 for PV enrichment. Can they explain why a discrepancy in PV isolation references this? Reference 74 does not mention anything related to PV isolation.

Response: We would like to sincerely apologise for this confusion which probably raised from our writing of this section. We would like to confirm that our PV isolation protocol is based on the published work of Real F protocol (reference 73). However, in the next section of the submitted manuscript, GC-MS analysis was described and that was performed based on protocol referenced in 74. In the revised manuscript, we have avoided this confusion and made correction by putting the references in the proper places. In the revised manuscript, this section (Line 663-678) read as,

“GC-MS analysis of LD-S and LD-R-PV

Following a 24Hrs infection period, KCs were harvested, washed with phosphate-buffered saline (PBS), and pelleted. Subsequent to this, PV isolation was carried out using the previously described protocol [35]. After PV isolation Bradford assay was carried out for normalizing the protein concentration. The resulting equal volume of PV pellet was suspended in 20 ml of dichloromethane: methanol (2:1, vol/vol) and incubated at 4°C for 24hours. After centrifugation (11,000 g, 1 hour, 4°C), the supernatant was checked through thin layer chromatography (TLC) and subsequently evaporated under vacuum. The residue and pellet were saponified with 30% potassium hydroxide (KOH) in methanol at 80°C for 2 hours. Sterols were extracted with n-hexane, evaporated, and dissolved in dichloromethane. A portion of the clear yellow sterol solution was treated with N, O-bis(trimethylsilyl)trifluoroacetamide (BSTFA) and heated at 80°C for 1 hour to form trimethylsilyl (TMS) ethers. Gas chromatography/mass spectrometry (GC/MS) analysis was performed using a Varian model 3400 chromatograph equipped with DB5 columns (methyl-phenylsiloxane ratio, 95/5; dimensions, 30 m by 0.25 mm). Helium was used as the gas carrier (1 ml/min). The column temperature was maintained at 270°C, with the injector and detector set at 300°C. A linear gradient from 150 to 180°C at 10°C/min was used for methyl esters, with MS conditions set at 280°C, 70 eV, and 2.2 kV[77].